# Disruption of the tumour-associated *EMP3* enhances erythroid proliferation and causes the MAM-negative phenotype

Nicole Thornton [1,11✉], Vanja Karamatic Crew [1,11], Louise Tilley[1,11], Carole A. Green[2], Chwen Ling Tay[1], Rebecca E. Griffiths [2], Belinda K. Singleton[2], Frances Spring[2], Piers Walser[1], Abdul Ghani Alattar [3], Benjamin Jones[1], Rosalind Laundy[1], Jill R. Storry [3,4], Mattias Möller [3], Lorna Wall [5], Richard Charlewood [5], Connie M. Westhoff[6], Christine Lomas-Francis[6], Vered Yahalom[7], Ute Feick[8], Axel Seltsam[9], Beate Mayer [10], Martin L. Olsson [3,4,12] & David J. Anstee[2,12]

The clinically important MAM blood group antigen is present on haematopoietic cells of all humans except rare MAM-negative individuals. Its molecular basis is unknown. By whole-exome sequencing we identify *EMP3*, encoding epithelial membrane protein 3 (EMP3), as a candidate gene, then demonstrate inactivating mutations in ten known MAM-negative individuals. We show that EMP3, a purported tumour suppressor in various solid tumours, is expressed in erythroid cells. Disruption of EMP3 by CRISPR/Cas9 gene editing in an immortalised human erythroid cell line (BEL-A2) abolishes MAM expression. We find EMP3 to associate with, and stabilise, CD44 in the plasma membrane. Furthermore, cultured erythroid progenitor cells from MAM-negative individuals show markedly increased proliferation and higher reticulocyte yields, suggesting an important regulatory role for EMP3 in erythropoiesis and control of cell production. Our data establish MAM as a new blood group system and demonstrate an interaction of EMP3 with the cell surface signalling molecule CD44.

[1] International Blood Group Reference Laboratory, NHS Blood and Transplant, Bristol, UK. [2] Bristol Institute for Transfusion Sciences, NHS Blood and Transplant and NIHR Blood and Transplant Unit in Red Cell Products, University of Bristol, Bristol, UK. [3] Division of Hematology and Transfusion Medicine, Department of Laboratory Medicine, Lund University, Lund, Sweden. [4] Department of Clinical Immunology and Transfusion Medicine, Office of Medical Services, Lund, Sweden. [5] Reference Laboratory, New Zealand Blood Service, Auckland, New Zealand. [6] New York Blood Centre, New York, USA. [7] Magen David Adom, National Blood Services, Ramat Gan, Israel. [8] Deutsches Rotes Kreuz, Blood Donor Service, Institute Bad Kreuznach, Bad Kreuznach, Germany. [9] German Red Cross Blood Service NSTOB, Institute Springe, Springe, Germany. [10] Institute of Transfusion Medicine, Charité-Universitätsmedizin Berlin, corporate member of Freie Universität Berlin, Humboldt-Universität zu Berlin, and Berlin Institute of Health, Berlin, Germany. [11] These authors contributed equally: Nicole Thornton, Vanja Karamatic Crew, Louise Tilley. [12] These authors jointly supervised this work: Martin L. Olsson, David J. Anstee. ✉email: nicole.thornton@nhsbt.nhs.uk

Blood group discovery has proven to be important, not only through enabling safe blood transfusion, but also by deepening our understanding of the red cell membrane and of other cells and tissues in health and disease[1–3]. Blood group antigens are carried by functional proteins, glycoproteins and glycolipids at the erythrocyte membrane surface and are defined by specific antibodies that recognise a polymorphism of a commonly expressed molecule. MAM is one of the few recognised high prevalence blood group antigens that has eluded molecular characterisation. The MAM-negative phenotype was first described in 1993, when an antibody to an unknown antigen of high prevalence was detected in the serum of a pregnant woman (M.A.M.) during routine antenatal testing. The infant was delivered prematurely following symptoms of foetal cardiac distress that were apparently unrelated to the antibody[4]. However, severe or even fatal haemolytic disease of the foetus and newborn has been caused by anti-MAM in subsequent cases (Supplementary Table 1).

Epithelial membrane protein 3 (EMP3) is a member of the peripheral myelin protein 22 (PMP22)/claudin superfamily of proteins, a protein family characterised by the members each having four transmembrane domains with two extracellular loops and one short intra-cellular loop[5]. Members of this family are involved in diverse cellular (mostly membrane) functions, ranging from membrane organisation to regulation of tight junctions[6]. Duplication of PMP22 is associated with a common form of hereditary autosomal dominant demyelinating neuropathy (Charcot–Marie–Tooth disease type 1A, CMT1A). Cultured fibroblasts from CMT1A patients with duplicated PMP22 have 1.5-fold elevated levels of PMP22 mRNA and this is accompanied by reduced mitotic potential and intracellular protein aggregates[7]. Point mutations in PMP22 are also causative of CMT1A and can result in truncated proteins retained within the cytosol[8]. Reconstitution of PMP22 into lipid vesicles results in the formation of myelin-like assemblies demonstrating that PMP22 plays a role in organising membrane ultrastructure[9]. Disease-causing variations have not been reported for EMP3 but a role in membrane assembly and proliferation would not be unexpected given what is known of PMP22. EMP3 has been reported as a tumour suppressor gene in various solid tumours and as a possible therapeutic target in cancer[10,11].

In this study, we set out to determine the genetic basis of the MAM-negative phenotype. We utilise comprehensive serological characterisation, followed by next generation sequencing and CRISPR/Cas9 gene editing to reveal the causative gene. Inactivating mutations in the EMP3 gene are identified in all ten known MAM-negative individuals. We observe a proliferation advantage in the ex vivo erythroid cell cultures of CD34+ progenitor cells of MAM-negative individuals and show clear association between EMP3 and CD44. This study demonstrates that EMP3 is expressed on erythrocytes and is the carrier molecule for the MAM antigen, establishing MAM as a new blood group system.

## Results

**Serological characterisation of MAM.** Comprehensive serological analysis of red cells from MAM-negative individuals showed normal expression of other high prevalence blood group antigens (Supplementary Table 2), except for antigens of the Indian blood group system, carried on CD44, which were expressed weakly (Supplementary Table 3, Supplementary Fig. 1). Although the reactivity of anti-MAM was not characteristic of a CD44-related antibody, this was the only indication of a potential association of MAM with a known red cell membrane protein and therefore this relationship was explored further in the monoclonal antibody-specific immobilisation of erythrocyte antigens (MAIEA) assay. The MAIEA immunoassay is primarily designed for locating blood group antigens on specific red cell membrane proteins[12]. Anti-MAM was tested in the MAIEA with 26 monoclonal antibodies, selected to target a total of 12 red cell membrane proteins including CD44 (Supplementary Table 4). Only the nine monoclonal antibodies specific for CD44 gave positive results with anti-MAM in the MAIEA, affirming a close, physical interaction between CD44 and MAM.

**Genetic analysis of MAM-negative phenotype.** Despite the serological evidence showing a clear link between CD44 and MAM, Sanger sequencing of the erythroid CD44 isoform, CD44H, showed no coding mutations in five known, unrelated, MAM-negative individuals (data not shown). To explore further and determine the genetic basis of the MAM-negative phenotype, DNA from these MAM-negative individuals was subjected to whole-exome sequence analysis. No coding mutations were observed in the entire CD44 gene, however, all five MAM-negative individuals had inactivating mutations in the gene encoding the transmembrane protein EMP3. Sanger sequencing of EMP3 confirmed the observed mutations in these five individuals and also demonstrated inactivating mutations in a further five MAM-negative individuals. The EMP3 mutations detected comprised whole gene deletion, single exonic deletions and a nonsense mutation; all predicted to abolish, or substantially alter, expression of EMP3 (Fig. 1, Supplementary Table 5). All nonsense mutations in EMP3 are rare (Supplementary Tables 6 and 7); of those, c.123 C > G (p.Tyr41Ter) is by far the most commonly encountered in the Genome Aggregation Database (gnomAD) (Supplementary Table 6), where it is present in 43 of 251,000 alleles (0.017%). The subjects in this study were not discovered by population frequency analysis, however, the c.123 C > G mutation was also the most common in our cohort, found in four propositae (two of them related).

**EMP3 is necessary for MAM expression.** Initial confirmation that EMP3 is the molecule responsible for expression of the MAM antigen was obtained by transduction with EMP3-specific shRNAs in the human erythroid progenitor cell line BEL-A2[13]. The undifferentiated wild-type BEL-A2 cells are predominantly in the erythroblast stage of development and express EMP3 at a low level, similar to glycophorin A and tetraspanin CD81 levels at the same stage (Supplementary Table 8). The EMP3-specific shRNA transduction resulted in a transient knockdown of EMP3, showing a reduction of MAM expression (Supplementary Fig. 2). The total ablation of EMP3 was achieved using CRISPR/Cas9[14,15] gene editing in the BEL-A2 cell line, where knockout (KO) cells showed completely abolished expression of MAM (Fig. 2, Supplementary Fig. 3). Subsequently, to provide further evidence of the link between EMP3 and MAM, overexpression of MAM was obtained in Daudi cells, which naturally have only low levels of EMP3, by transfection of wild-type EMP3 (Fig. 3). Through these three different experimental approaches, we have consistently demonstrated that MAM expression is reliant on intact EMP3. Monoclonal anti-EMP3 reagents were found to perform generally poorly by a number of methods, including serological testing with untreated red cells. However, one anti-EMP3 reagent gave positive reactions with enzyme treated red cells of two MAM-positive controls, whilst failing to react with red cells of MAM-negative patients treated in the same way (Supplementary Table 9) thereby demonstrating binding of anti-EMP3 to MAM. Subsequent antibody inhibition studies confirmed that anti-MAM specifically blocks the binding of EMP3 with anti-EMP3 (Supplementary Table 10). Enzyme treatment of red cells often enhances the reactivity of the antibodies directed towards epitopes located close

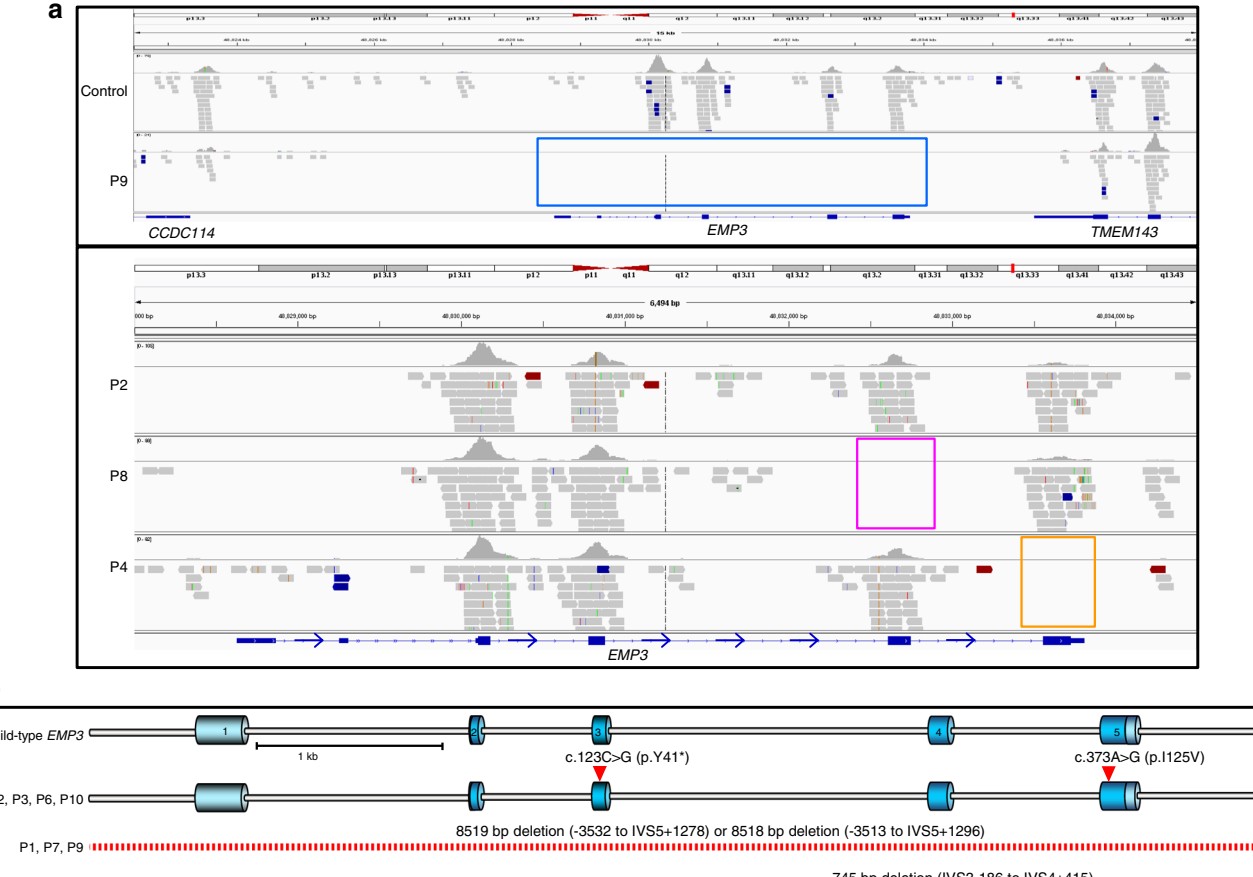

**Fig. 1 DNA sequencing of ten unrelated MAM-negative individuals revealed inactivating mutations in *EMP3*. a** Exome sequence alignments over the *EMP3* gene region demonstrate four inactivating mutations in five MAM-negative individuals. *EMP3* was the only candidate gene that passed our filtering strategy with predicted loss of function mutations found in all tested MAM-negative samples. Upper panel (chr19; 48,822,471 to 48,837,471) shows a complete deletion of *EMP3* in P9 (blue box) revealed by a lack of coverage over any targeted exons as compared to control, although flanking genes *CCDC114* and *TMEM143* show sequencing coverage. Lower panel (chr19; 48,828,000 to 48,834,500) shows homozygous nonsense mutation (c.123C > G; p.Tyr41Ter) in P2 (brown line); deletion of *EMP3* exon 4 in P8 (pink box) and P5 (data not shown); deletion of *EMP3* exon 5 in P4 (orange box). **b** Sanger sequencing was used to confirm these mutations and identify *EMP3* mutations in a further five MAM-negative individuals and deletion breakpoints were identified where appropriate (summarised in Supplementary Table 5). *EMP3* inactivating mutations in all patients (P1–P10) are depicted on the gene schematic, with dark blue areas depicting coding exons, light blue areas UTRs and deleted regions depicted by dashed red lines.

to the plasma membrane, as our model suggests is the case with EMP3 (Fig. 4).

**MAM expression on platelets**. EMP3 has been reported to be widely expressed in diverse human tissues (Supplementary Fig. 4), including a wide variety of haematopoietic tissues and cell types[16]. Conflicting data regarding MAM expression on platelets[17], coupled with reports of foetal and newborn low platelet counts in infants of MAM-negative individuals (P1, P2, P9; data not shown) prompted us to investigate MAM expression on platelets. We found that MAM is expressed on platelets and MAM expression is independent of the activation status of platelets (Supplementary Fig. 5).

**EMP3 effect on cell proliferation in erythroid cell culture**. The function of EMP3 currently still lacks definition; *EMP3* has been reported as both a tumour suppressor gene and an oncogene in studies of cancers from diverse tissues (Supplementary Table 11). Due to this apparently disparate published data, we endeavoured

to examine the role of EMP3 in erythropoiesis using an in vitro culture system[18]. CD34+ cells from peripheral blood of two MAM-negative individuals and four age- and gender-matched MAM-positive controls were cultured and compared under explicit expansion and differentiation conditions. The results revealed a marked increase in proliferation (average of 5-fold on day 21 of culture) of MAM-negative cultures, suggesting EMP3 acts as a suppressor of proliferation in normal erythropoiesis (Fig. 5, Supplementary Fig. 6). However, MAM-negative individuals do not exhibit any apparent erythrocytosis or reticulocytosis (data not shown), suggesting factors external to the developing red cell or compensatory mechanisms influence erythroid proliferation in vivo.

**EMP3 interactions with CD44**. Our serological testing and flow cytometry analysis suggested a relationship between the MAM-negative phenotype and reduced CD44 expression (see Supplementary Table 3, Supplementary Fig. 1). Immunoblotting confirmed the presence of reduced CD44 on erythrocytes of

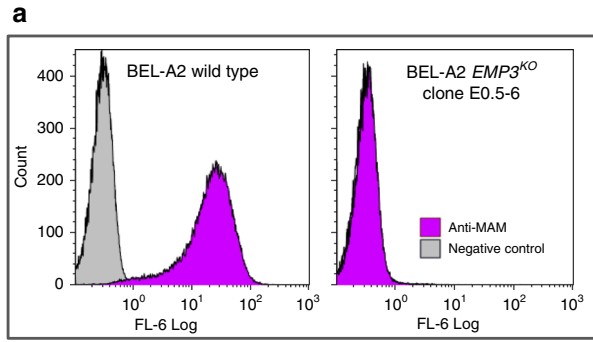

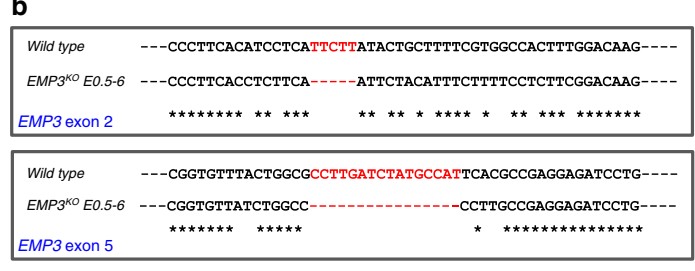

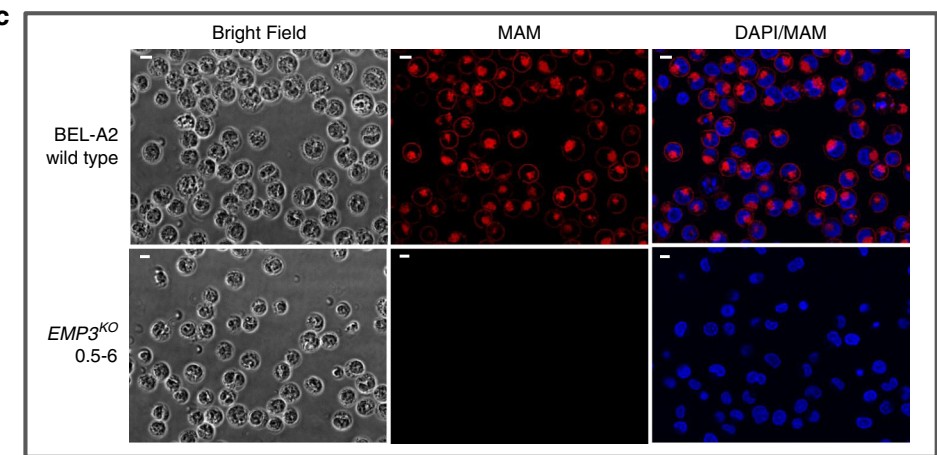

**Fig. 2 Disruption of the *EMP3* gene in BEL-A2 cell line demonstrated its underlying role in MAM antigen expression. a** Flow cytometry analysis of *EMP3*$^{KO}$ clone E0.5–6 with anti-MAM (P9 eluate). The CRISPR/Cas9-mediated gene editing system was used to disrupt *EMP3* in BEL-A2 immortalised erythroid cell line. A total of eight transgenic BEL-A2 transfectant clones had no detectable MAM on cell surfaces, revealing the *EMP3* gene to be responsible for MAM expression. **b** Sanger sequencing of *EMP3* exons 2 and 5 for *EMP3*$^{KO}$ clone E0.5–6. Sanger sequencing confirmed inactivating deletions in all KO clones, in *EMP3* exon 2 and exon 5. In clone E0.5–6, deletions NM_001425.2: c.44_48del in exon 2 and NM_001425.2: c.344_359del in exon 5 were close to the location of guide DNAs and both would have introduced a reading frame shift and aberrant termination of the protein translation (p.Ile15AsnfsTer40 and p.Ala115ValfsTer176, respectively). **c** Immunofluorescence assay with BEL-A2 *EMP3*$^{KO}$ clone E0.5–6. Four KO clones were stained with anti-MAM (P9 eluate; shown in red), with duplication, for immunofluorescence assays (magnification ×40) to show a complete loss of MAM on their cell surfaces in all clones. DAPI staining shown in blue. Scale bars are 10 μm. Several KO clones were also stained with commercially available anti-EMP3, as shown in Supplementary Fig. 3.

MAM-negative individuals, and subsequent immunoprecipitation provided evidence of a direct association between MAM and CD44 in normal MAM-positive red cells (Supplementary Fig. 7). In addition, proteomic analysis of immunoprecipitated erythrocyte-derived CD44 also confirmed the association between CD44 and EMP3 (Supplementary Proteomics Data File 1) as previously indicated by the results of the MAIEA assay (see Supplementary Table 4).

Based on this clear link, the impact of EMP3 ablation on the expression of CD44 was further explored by confocal microscopy of cultured erythroid cells.

CD44 has been the focus of many studies in relation to cancer[19,20]. It is a type 1 membrane protein exhibiting multiple isoforms in different tissues. All isoforms connect the extra-cellular matrix (notably through interaction with hyaluronan) with the membrane cytoskeleton and the intracellular milieu, thereby acting as a signalling platform known to modulate cell proliferation (reviewed in Thorne et al.)[21]. When normal adult erythroblasts were produced by in vitro culture, CD44 was preferentially located at the cleavage furrow of dividing cells (Fig. 6a) along with myosin IIb, actin, CD147 and the tetraspanins CD63, CD81 and CD82. When dividing erythroid progenitors from a MAM-negative individual were examined, strong CD44 staining at the cleavage furrow was still apparent,

but in addition, CD44 staining was observed over the rest of the cell surface (Fig. 6a, b). No difference in expression or localisation of myosin IIb, CD147, CD63, CD81 or CD82 was observed between the control and MAM-negative cells (data not shown). Numerous CD44-positive vesicles co-localising particularly with CD81 (Fig. 6c), and to a lesser extent with CD82 and CD63 (data not shown), were also observed in late stage erythroblasts cultured from a MAM-negative individual but not in comparable cells from MAM-positive individuals. Our data suggest EMP3 may have a function in regulating the level of CD44 at the cell surface of erythroid progenitors. This is analogous to the well-established role of CD81 in the regulation of CD19 expression at the surface of B lymphocytes[22]. Therefore, this raises the possibility that in the absence of EMP3, the greater levels and broad distribution of CD44 we observed in MAM-negative proerythroblasts, alter the dynamics of the actinomyosin function at the cleavage furrow[23], thereby increasing the cell proliferation rate. Despite more abundant CD44 in early MAM-negative cultured erythroblasts, mature erythrocytes and cultured reticulocytes of MAM-negative individuals have a gross reduction of CD44 expression (Supplementary Fig. 8). Most CD44 appears to be lost with the nucleus during enucleation in the final stages of erythrocyte maturation (Fig. 6d). We can speculate that in the absence of the stabilising effect of EMP3 in MAM-negative cells,

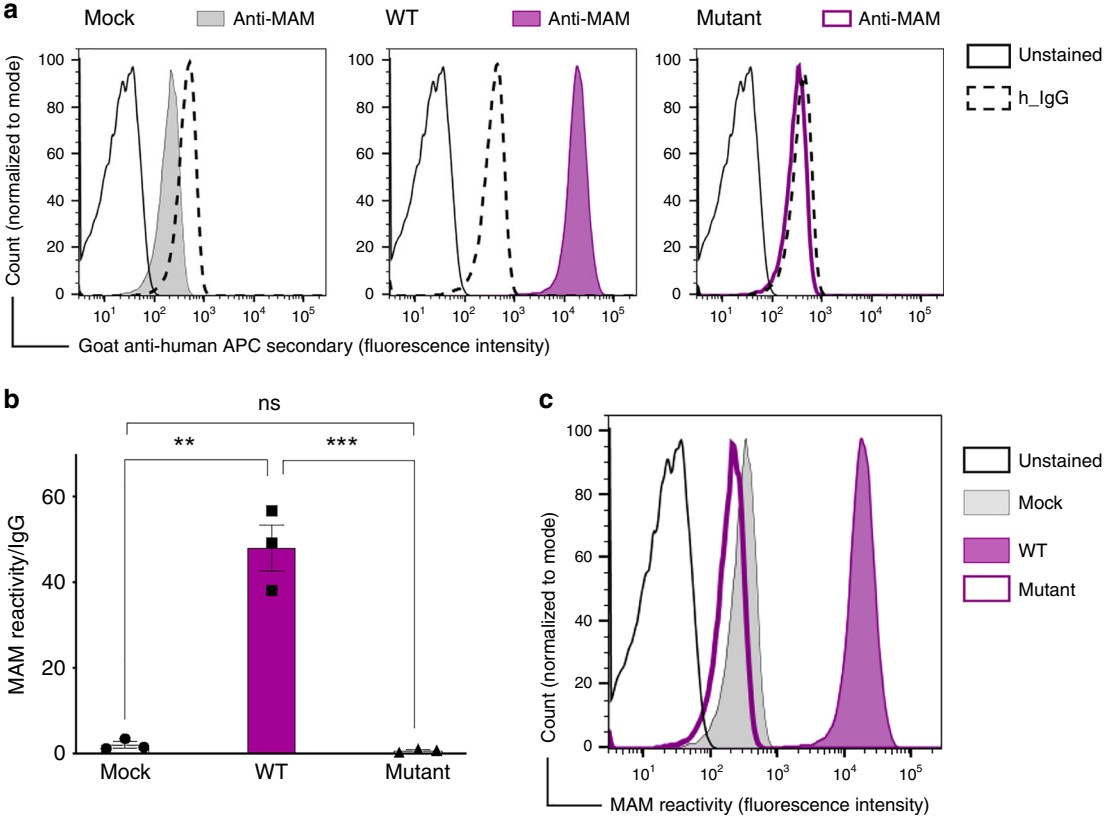

**Fig. 3 EMP3 transfection causes MAM expression in Daudi cells. a** Flow cytometry analysis showing establishment of MAM expression in Daudi cells. The representative histogram shows IgG (negative control) and MAM reactivity following transfection of Daudi cells with either pEF1α-IRES-ZsGreen1 empty vector (mock), EMP3 wild-type (WT) or EMP3 mutant containing a premature stop codon (c.123 C > G, p.Tyr41Ter). **b** Overexpression of EMP3 in Daudi cells shown by mean fluorescent intensity from three independent experiments; MAM expression is shown as a ratio between MAM reactivity and isotype control (IgG) reactivity in Daudi cells transfected by either mock, EMP3 WT or EMP3 mutant vectors. **P = 0.0011, ***P = 0.001 (two-tailed unpaired student *t* test). Error bars represent standard error of the mean (SEM). Source data are provided as a Source Data file. **c** The histogram shows MAM reactivity following transfection of Daudi cells with either mock, wild-type (WT) or EMP3 mutant compared to unstained cells.

CD44 may also be lost via a different mechanism involving plasma membrane vesiculation (see Fig. 6c).

To further investigate the interdependency of MAM and CD44, a CD44 KO BEL-A2 cell line was developed (Supplementary Fig. 9a) and tested by flow cytometry for MAM expression. Expression of MAM was clearly maintained, even in the absence of CD44 (Supplementary Fig. 9b). This suggests that, although EMP3 appears to have a mechanistic role in stabilisation of CD44 in the plasma membrane, this relationship is not reciprocal, since MAM is expressed independently of CD44.

## Discussion

We have demonstrated that the rare but clinically important MAM-negative phenotype is caused by inactivating mutations in the *EMP3* gene, thereby establishing the molecular background of one of the few remaining genetically unresolved blood group antigens. While MAM-negative individuals readily produce anti-MAM when challenged, and the clinical significance of this antibody in pregnancy is well established, there is no apparent disease associated with the absence of EMP3. Despite the putative cancer association of EMP3, we have no evidence that EMP3null individuals have an increased risk of cancer. Rather, our results suggest a role for EMP3 in stabilising CD44 in the erythroblast plasma membrane. EMP3 may regulate formation of the signalling complex by controlling incorporation of CD44 into lipid rafts. Despite a clear association between CD44 and MAM, demonstrated by both MAIEA and immunoprecipitation, our

evidence suggests EMP3 expression during biosynthesis is independent of CD44 expression. *CD44* mRNA expression levels remained unchanged when EMP3 expression was knocked down in BEL-A2 cells (see Supplementary Fig. 2a) and CD44 expression on MAM-negative and control cells appeared similar during the early stages of cell expansion in ex vivo cell cultures (see Supplementary Fig. 8b). It is therefore not surprising that EMP3 levels were not affected in the CD44 KO BEL-A2 cell line.

The external environment of developing erythroid cells found in bone marrow is not replicated in our in vitro erythroid culture system. It does not contain extracellular matrix proteins that could affect signalling through CD44, or bone marrow macrophages which are known to have multiple interactions, some of which affect erythroid proliferation in erythroblastic islands[24]. The absence of some or all of these extracellular signals may explain why the enhanced erythroid proliferation we observe in cultures from MAM-negative individuals does not appear to be replicated in vivo. Since in vivo erythropoietin levels decrease when oxygen saturation is sufficient, and the lack of EMP3 is unlikely to render cells erythropoietin independent, we can speculate that the erythropoietin concentration will override the importance of EMP3 regulation in vivo, or other compensatory mechanisms may be at play. Loss of EMP3:CD44 interaction has also been reported in glioma, where ablation of EMP3 attenuates proliferation[25]. The apparently conflicting effects of EMP3 on proliferation of cancer cells in different tissues (see Supplementary Table 11) may simply reflect the different environments, both internal and external, that interact with the EMP3:CD44 membrane bound signalling complex.

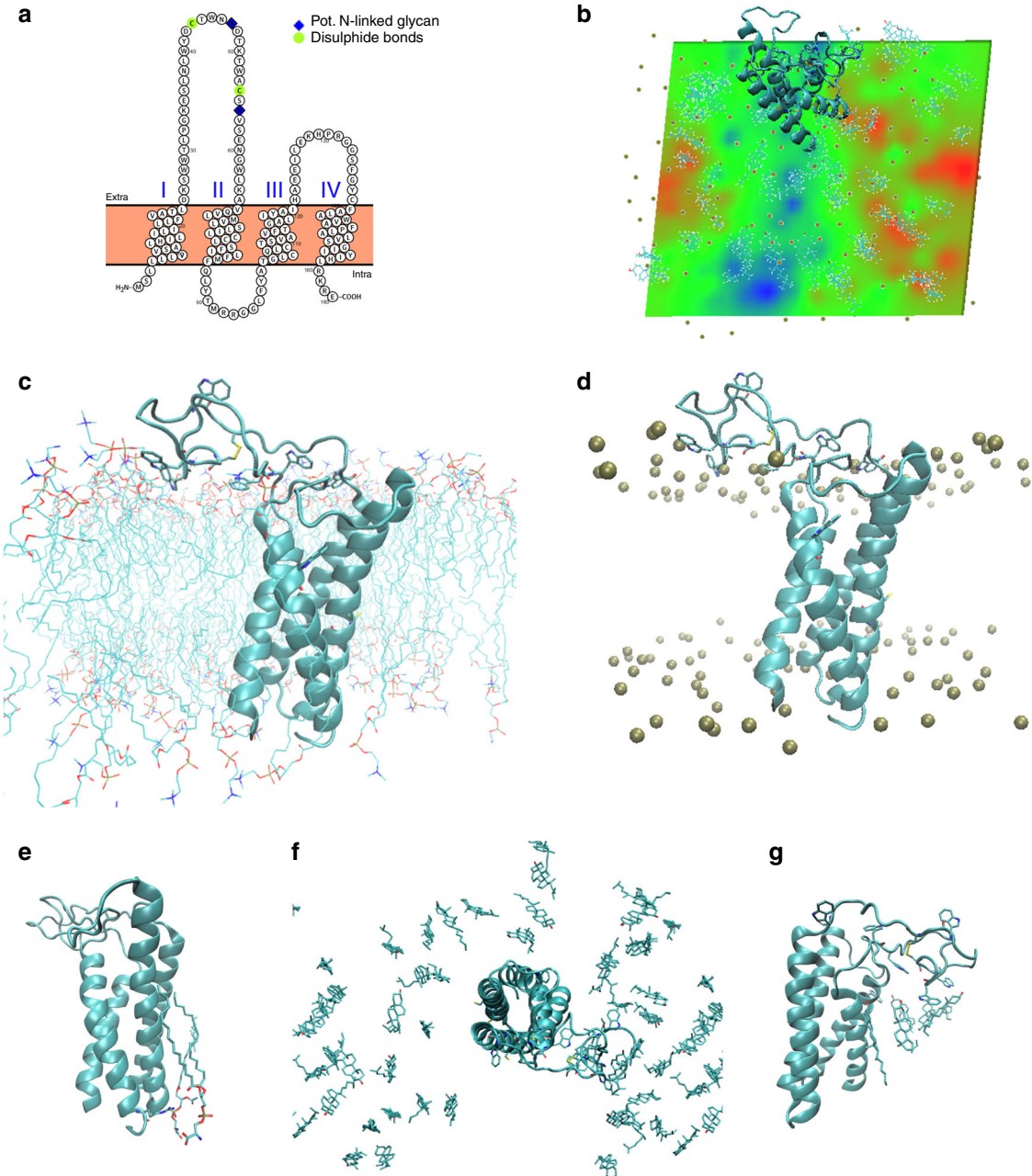

**Fig. 4 Structural calculations and predications on membrane embedded EMP3. a** Homology modelling using PROTTER software. Proposed topology of EMP3 based on homology with claudin-19. I–IV represent transmembrane segments with both termini placed in the cytoplasm. Proposed posttranslational modification sites are indicated. Model similar to previously published EMP family model[35]. **b** Predicted effects of first extracellular loop interactions with the membrane interface and consequence for apparent membrane thickness. The first extracellular loop was remodelled from initial homology model (based on claudin-19, PDB ID 3 × 29)[34] and embedded in a representative cytoplasmic membrane composed of (mole percentages) 50% phosphatidylcholine/25% cholesterol/15% phosphatidylethanolamine/10% phosphatidylserine (inner leaflet only). All-atom explicit solvent molecular dynamics (MD) calculations (150 ns calculation time) were performed to refine initial homology model. Membrane thickness (phosphate-to-phosphate) is represented as a heat map with relative changes in bilayer dimensions colour-coded (green: average; red: increased; blue: reduced). The polypeptide is represented as a cyan ribbon; gold spheres indicate phosphate atoms of phospholipids. N.B. due to periodicity of the calculation space, the larger blue region at bottom of map lies beneath the large extracellular loop of EMP3 (at top of map extending over the edge). **c** Tentative model after MD calculations. EMP3 is represented as cyan ribbon cartoon and phospholipids as stick representations. Tryptophan residues in the first extracellular loop are shown (stick representation) to illustrate the high density of tryptophan residues within this loop (frequency of 0.15) as well as assumed intra-molecular disulphide bond (yellow). **d** as **c** but with phosphate atoms represented by gold spheres. **e** Phosphatidylserine molecules stably associated with the transmembrane segment are indicated by stick representation. These were observed to be coordinated by Arg160 (indicated by stick representation) located at the membrane interface near the C-terminus of the protein and remained associated with Arg160 for the majority of the trajectory. **f** Distribution of cholesterol molecules after MD calculations (stick representation), illustrating the relative accumulation of cholesterol beneath first extracellular loop (blue region in panel B/reduced membrane thickness). View from top of the membrane block. **g** Close-up of cholesterol accumulation beneath first extracellular loop as shown in (**f**), viewed from side.

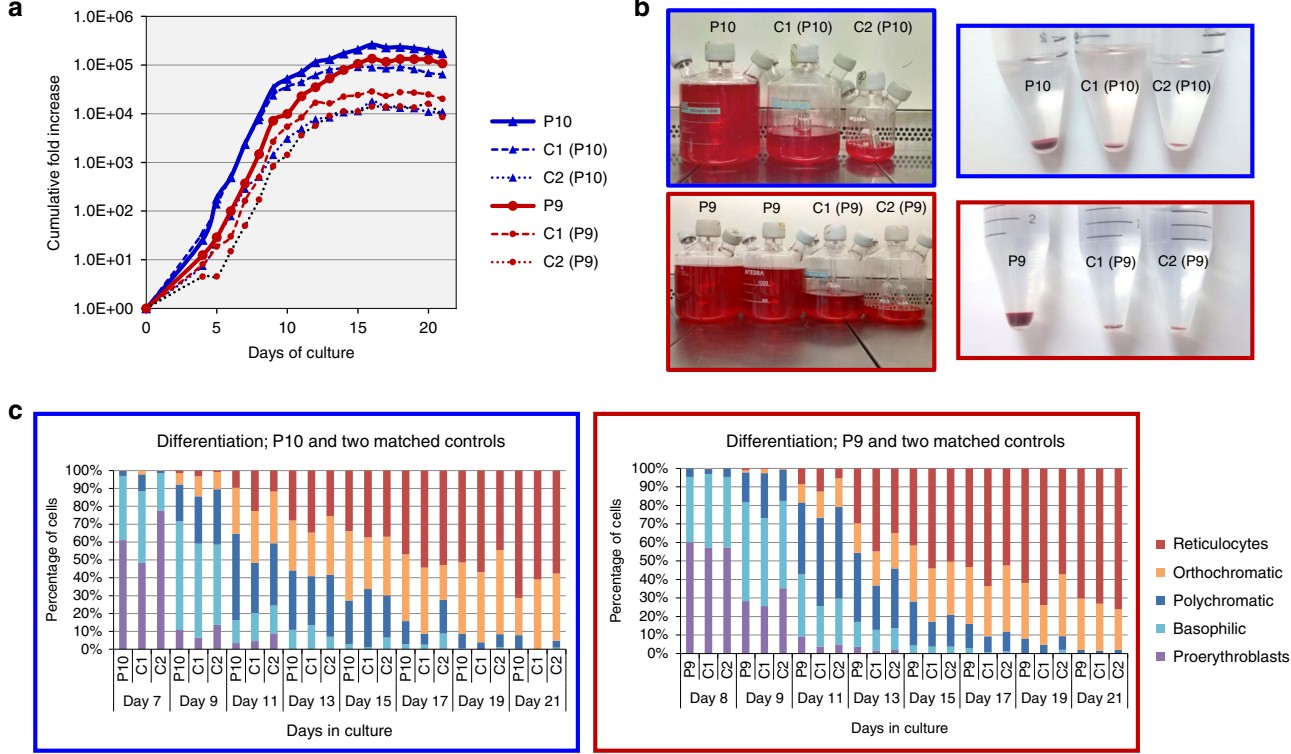

**Fig. 5 CD34+ cells isolated from two MAM-negative whole blood samples, P9 (red) and P10 (blue) and cultured under explicit proliferation conditions following the three-stage protocol described by Griffiths et al.[18] in two ex vivo cell culture experiments showed enhanced erythroid proliferation compared to their respective age and gender-matched controls (C1 and C2). a** Standardised cumulative fold proliferation of erythroid cells in cultures. In each experiment, all samples were maintained at equal cell densities, with passaging carried out as necessary. MAM-negative cultures showed greater expansion compared to the averaged expansion of their respective controls (7.5-fold greater in P9 and 4.62-fold greater in P10 on day 21). Proliferation data from additional cell culture experiments with P9 and P10 supported these findings and are presented in Supplementary Fig. 6. Source data are provided as a Source Data file. **b** As cell cultures were adjusted to maintain equal cell densities, markedly different final volumes were obtained at the end of the culture. On day 21, enucleation rates of both MAM-negative and control samples were approximately 70%. All samples (total volumes) were filtered using a standardised filtration protocol. Purity of resulting reticulocyte population was >99.2% in all samples. Comparison of packed cell pellets of filtered reticulocytes reflected the greater proliferation capabilities observed in the MAM-negative cultures. **c** The morphological profiles of the MAM-negative and matched control cells at different stages of erythroid cultures were very similar. No significant differences in the erythroid cell types between MAM-negative and matched controls were observed, with the exception of P9 and P10 reticulocytes (ANOVA repeat measures; $P = 0.01$), suggesting that the prevalence of each cell type remained constant between the cultures. Therefore, the enhanced MAM-negative cell proliferation was not the result of imbalanced or altered erythroid differentiation. Source data are provided as a Source Data file.

Our data are consistent with a role for EMP3 in the membrane assembly of a signalling complex with CD44 in the plasma membrane of erythroid cells which participates in the regulation of cellular proliferation. There is considerable interest in generating therapeutic quantities of red cells in vitro for patients to whom blood group compatible blood is difficult or impossible to source (reviewed in Severn and Toye)[26]. Further elucidation of the EMP3:CD44 complex in erythroid cells may provide a route to manipulation of ex vivo erythroid cultures to optimise yield of red cell product. Furthermore, clarification of the composition and role of the EMP3:CD44 complex in other cells and tissues offers an approach to understanding and manipulating the tumour suppressor and oncogenic properties of EMP3 in diverse cancers.

## Methods

**Subjects**. All blood samples were procured and the study was conducted according to NHS Blood and Transplant (NHSBT) Research and Development governance requirements and ethical standards. Approval for the study was granted by NHS Health Research Authority, Research Ethics Committee reference 12/SW/0199. Subjects from whom blood samples were taken for the purpose of this study gave their written informed consent. Samples from 10 MAM-negative individuals, all with anti-MAM, were obtained from cryopreserved rare reference material collections from either IBGRL ($n = 7$) or NYBC ($n = 3$) as outlined in Supplementary

Table 1. Additional rare phenotype red blood cells (RBCs) and antisera were obtained from the IBGRL rare reference collection. Ethylenediaminetetraacetic acid (EDTA) peripheral blood samples from voluntary NHSBT blood donors, who had consented to the use of their blood for research purposes, were used as MAM-positive controls. Fresh blood samples were obtained from patient P9 and family members; P9-mother, P9-father and P9-sibling, following informed consent. P10 is a blood donor and waste products (EDTA whole blood tubes and citrated buffy coats from waste bags produced in the Reveos blood component system) from regular blood donations were obtained from this donor and age- and gender-matched control donors following approval by the advisory board at the Department of Clinical Immunology and Transfusion Medicine in Lund, Sweden. All materials were anonymized at the blood centre.

**Enzyme treatment and chemical modification of RBCs**. RBCs were treated with papain (NHSBT Reagents) and peptide-N-glycosidase-F (PNGase F; New England Biolabs). All incubations were carried out at 37 °C. For papain treatment, one volume of washed packed RBCs was incubated with two volumes of papain for 3 min, according to manufacturer's instructions. For PNGase F treatment, aliquots of 200 μl of washed packed RBCs were incubated >14 h in 300 μl of treatment buffer containing PNGase F, as per manufacturer's instructions. Following all treatment incubations, RBCs were washed a minimum of four times with phosphate-buffered saline (PBS) until the wash supernatant was clear.

**Serological testing**. Standard agglutination techniques were used for assessment of anti-MAM reactivity, RBC phenotyping and blocking tests[27]. For the indirect antiglobulin test (IAT), a low-ionic strength saline tube method was used, where the secondary antibody was anti-human globulin (Millipore) polyspecific for

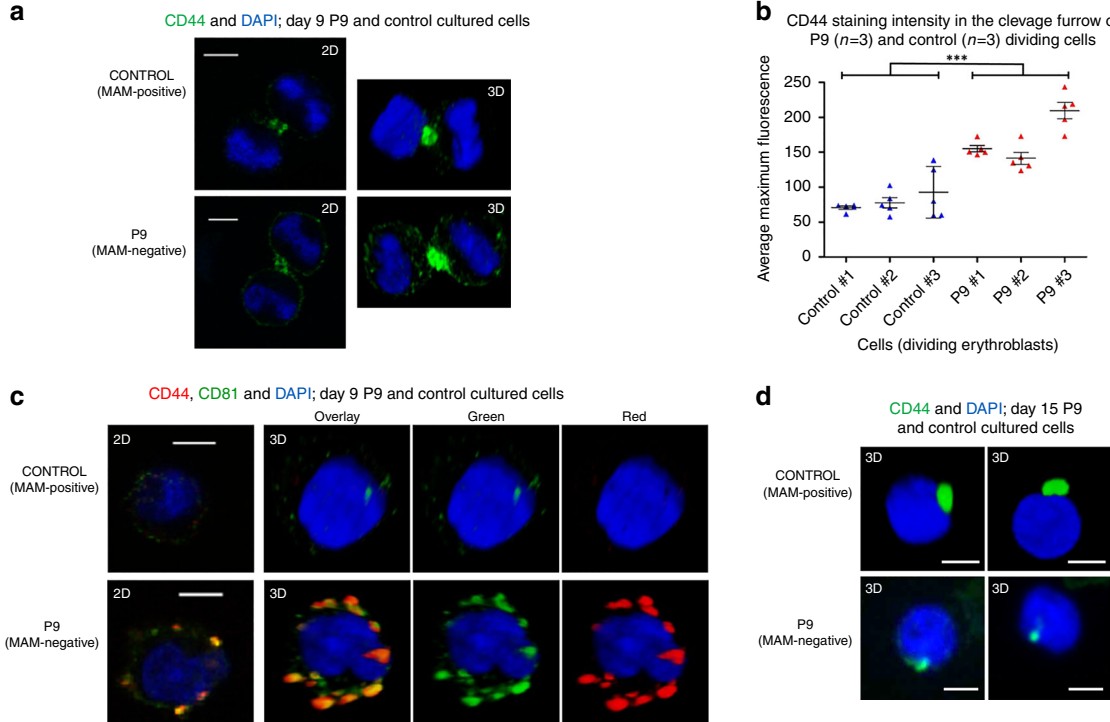

**Fig. 6 Confocal microscope analysis of MAM-negative P9, MAM-positive control erythroblasts and enucleating erythroblasts cultured as described in Supplementary Fig. 6.** Control MAM-positive and P9 MAM-negative erythroblasts were harvested on day 9 and 15, fixed in 1% (wt/vol) paraformaldehyde, permeabilised with 0.05% (wt/vol) saponin and stained for the presence of CD44 and tetraspanin CD81. Nuclei were stained with DAPI (blue). Representative cells shown from approximately 10 micrographs per sampling day. **a** Dividing erythroblasts were stained with anti-CD44 (green) and shown in 2D and 3D. Scale bars are 5 μm. **b** Average CD44 staining intensities ($n = 5$ independent measurements from each dividing cell) of three P9 and three control dividing cells were obtained with Leica LAS AF software. MAM-negative samples showed significantly larger maximum fluorescence in the cleavage furrows than the control samples (two-tailed $t$ test; $df = 28$, $P = 1.14 \times 10^{-8}$). Data are presented as mean values, with error bars representing standard deviation. Source data are provided as a Source Data file. **c** Erythroblasts stained with anti-CD44 (red) and anti-CD81 (green). Shown in 2D and 3D. Scale bars are 5 μm. **d** Representative free nuclei stained with anti-CD44 (green) showing CD44 to be associated with free nuclei after enucleation in the final stages of erythrocyte maturation. Shown in 3D. Scale bars are 5 μm.

human primary antibodies (in-house IBGRL reference collection), or polyclonal rabbit anti-mouse immunoglobulin G (IgG; Dako) for mouse primary monoclonal antibodies (IBGRL Research Products and Japanese Red Cross). Anti-MAM eluates were prepared using Gamma ELU-KIT™ II rapid acid elution kit (Immucor). Agglutination was scored on a scale of 0 (negative) to 5+ (strongest positive).

For testing with monoclonal anti-EMP3, RBCs from P8, P9 and two MAM-positive controls were papain treated, followed by PGNaseF treatment and then tested in parallel with anti-MAM and monoclonal anti-EMP3 (clone 439949; R&D Systems). The papain + PNGaseF (two-stage) treated cells were also tested with anti-Rh (BRIC69; IBGRL Research Products) and anti-RhAG (LA1818; IBGRL Research Products) to check the treatment efficacy and specificity. For blocking tests, 5 μl washed, packed, papain + PNGase F (two-stage) treated RBCs were incubated first with 10 μl human anti-MAM eluate for 1 h at 37 °C, washed four times, and incubated with mouse monoclonal anti-EMP3 (clone 439949), for 1 h at room temperature, then subjected to an IAT using anti-mouse globulin. An unrelated murine monoclonal antibody anti-Rh (BRIC69) was tested in parallel. An example of human polyclonal anti-Jkᵇ (Lorne Laboratories), was used to sensitise control cells in order to demonstrate specific blocking.

**MAIEA assay.** The monoclonal antibody-specific immobilisation of erythrocyte antigens (MAIEA) enzyme-linked immunoadsorbant assay was carried out as described by Petty et al.[13] using both MAM-positive and MAM-negative RBCs, incubated with human anti-MAM and different mouse monoclonal IgG antibodies of known specificity [BRIC222, BRIC241, BRIC205, BRIC214, BRIC219, BRIC223, BRIC225, BRIC235, KZ1, BRIC224, BRIC221, BRIC18, BRIC203, LA1818, AE-1, BRIC230, BRIC110, BRIC5, BRIC229, BRIC6, BIRMA84b, BRIC125 (IBGRL Research Products), HIM6 (BioLegend), UM-8D6, MEM-M6/6 (Abcam) and MIMA123 (NYBC)] in consecutive 1 h incubations at 37 °C. The sensitised RBCs were lysed and membranes solubilised using a Triton-X based solubilising buffer. Trimolecular complexes of the protein-bound human and mouse antibodies were captured in flat bottomed microplate wells coated with Fcγ fragment-specific goat anti-mouse IgG (Jackson ImmunoResearch Europe) diluted 1:1000. Immobilised complexes were detected by Fcγ fragment-specific goat anti-human peroxidase

conjugated IgG (Stratech Scientific Ltd.). Lysates were tested in triplicate and the mean adsorbance calculated. For each MAb tested, the mean adsorbance of the MAM-positive sample was expressed as a ratio of the mean adsorbance of the MAM-negative sample, to account for non-specific binding and background adsorbance. An adsorbance ratio of greater than 2.0 was designated a positive result[12]. Human derived antibodies (IBGRL reference collection) of known specificity were used as positive controls on each microplate to control for technique failure.

**DNA preparation.** Genomic DNA (gDNA) was extracted from residual buffy coat material contained within the archived MAM-negative RBC samples and whole blood of additional MAM-negative samples and family members, using a commercial genomic DNA isolation kit according to manufacturer instructions (QIAamp DNA blood mini kit; Qiagen). Quantity of extracted gDNA was measured using a commercial fluorescence-based quantitation assay kit (Quant-iT dsDNA BR assay kit; Invitrogen).

**Next generation sequencing library preparation.** Library preparation, indexing and target enrichment for exome sequencing was carried out using the Nextera Rapid Capture Exome kit (Illumina), targeting 45 Mb of exonic coding sequence. Paired-end (2 × 87 cycle) single-plex sequencing was carried out on a MiSeq instrument (Illumina). Secondary data analysis, including alignment of reads against human reference genome hg19, was carried out using MiSeq Reporter v2.5.1 (Illumina) and variants were called and annotated using Illumina Variant Studio v2.2. Homozygous or compound heterozygous variants with allele frequency <1%, predicted to result in loss of protein function, were considered as potential causes of the MAM-negative phenotype. Selected alignments were further visualised using Integrative Genomics Viewer (IGV v2.3)[28,29].

**Genetic analysis of EMP3 and CD44H.** *EMP3* coding exons (2–5) and *CD44H* coding exons present in the erythroid isoform (1–5 and 15–17) were amplified by polymerase chain reaction (PCR) using intronic primers flanking the relevant exon (Eurofins Genomics). For primer sequences and PCR conditions see

Supplementary Table 12. PCR reactions were carried out using BioTaq polymerase (Bioline), AmpliTaq Gold (Applied Biosystems), or Expand High Fidelity PCR system (Sigma-Aldrich), under manufacturers' recommended conditions. PCR products were visualised by agarose gel electrophoresis and prepared for sequencing following commercial PCR clean-up protocols for SureClean (Bioline) or GenElute (Sigma-Aldrich). PCR products were subjected to bidirectional direct Sanger sequencing with forward and reverse PCR primers using a capillary automated DNA sequencer (3130xL Genetic Analyser; Applied Biosystems). SeqScape software v.3 (Applied Biosystems) was used to align all exon sequences to *Homo sapiens EMP3* (NC_000019.10) or *CD44* (NG_008937.1) reference sequence for comparative analysis and detection of gene mutations. Analysis of deletions involving *EMP3* exon 4, exon 5 and the complete *EMP3* gene was carried out by PCR amplification of products spanning the deleted regions (see Supplementary Table 12). PCR products from individuals harbouring deletions were discriminated from larger wild-type products by agarose gel electrophoresis. PCR products were prepared for sequencing with Affymetrix ExoSAP-IT (Thermo Fisher Scientific) and sequenced with forward and reverse PCR primers, or internal sequencing primers, as shown in Supplementary Table 12.

**Flow cytometry**. Erythrocytes, BEL-A2 cells and cultured erythroid progenitor cells were analysed by flow cytometry after labelling with the following antibodies: mouse monoclonal anti-EMP3 (R&D Systems) and anti-CD44 (BRIC222 and BRIC235; IBGRL Research Products). Isotype controls (BioRad) were tested in parallel to ascertain background fluorescence. For detection of MAM, human anti-MAM eluates were prepared by adsorption and elution from antigen-positive cells (Elu-Kit II; Gamma Biologicals). In all experiments, cells were incubated with primary antibody at room temperature for 30 min and washed once. Bound antibody was detected by adding fluorescein isothiocyanate or R-phycoerythrin rabbit anti–mouse globulins (Dako), or Alexa Fluor® 647-AffiniPure F(ab')2 fragment goat anti-human IgG (Jackson ImmunoResearch Europe) and incubated at 4 °C for 30 min. Cells were washed once, and the fluorescence geometric mean was recorded for each test. Flow cytometry was performed on a Navios instrument (Beckman Coulter) and analysed by Kaluza v.1.2 and v2.1 (Beckman Coulter). Gating strategy used in flow cytometry experiments is shown in Supplementary Fig. 10.

**Lentivirus preparation and shRNA BEL-A2 cell transduction**. HEK293T cells (Takara Bio Europe) were seeded ($8 \times 10^6$) in T75 static tissue culture flasks 24 h prior to transfection with 15 µg psPAX2 (packaging plasmid), 5 µg pMDG2 (envelope plasmid) and 20 µg shRNA plasmid (GeneCopeia Inc.) using polyethylenimine (PEI). *EMP3* clone set #HSH004823-LVRU6GP (target sequences sh1: 5′-GAGAGAAGAGACGCAGAAGGA-3′, sh2: 5′-AGAGTCCCTGAATCTC TGGTA-3′, sh3: 5′-TGCAGTAATGTCAGCGAGAAT-3′, sh4: 5′-ATCCTCATTC TTATACTGCTT-3′) or a scrambled control #CSHCTR001-1-LVRU6GP (target sequence 5′-GCTTCGCGCCGTAGTCTTA-3′) were used. The PEI/DNA complexes were incubated with the cells at 37 °C for 4 h, then replaced with fresh medium. Viral supernatants were harvested after 48 and 72 h, pooled, filtered and concentrated using Lenti-X Concentrator (Takara Bio Europe). Viral pellets were resuspended in 1.5 ml StemSpan SFEM (STEMCELL Technologies). A total of $1 \times 10^6$ BEL-A2 cells were transduced with 0.25–0.5 ml concentrated virus in the presence of 8 µg per ml Polybrene (Sigma-Aldrich) for 1 h, when 5 ml of expansion medium were added[13]. After 48 h, cells were washed in Hanks' Balanced Salt Solution (Sigma-Aldrich) and seeded in fresh expansion medium. Initial samples were taken for flow cytometry to assess transduction efficiency, measured by green fluorescent protein (GFP) expression, followed by puromycin selection (0.5 µg per ml; Sigma-Aldrich) for at least 48 h. Cells were sampled on days 2 or 3, and on days 9 or 10 post-transduction for storage at −80 °C in RNAlater (Thermo Fisher Scientific). Expression of EMP3 was assessed by flow cytometry and confocal microscopy nine or more days post-transduction.

**qPCR of BEL-A2 cells transduced with *EMP3*-targeting shRNAs**. Total RNA was extracted from transduced BEL-A2 cells stored in RNAlater, cDNA was prepared and quantitative real-time PCR (qPCR) was used to quantify expression of *EMP3* and *CD44H* RNA, as described in Singleton et al.[30] with some modifications. Target gene expression was normalised to the geometric mean of two reference genes and compared with a calibrator sample prepared from untransduced BEL-A2 cells. The following TaqMan™ Gene Expression assays were used: *EMP3* (Hs00171319_m1), *PABPC1* (Hs00743792_s1) and *UBC* (Hs01871556_s1) (all Applied Biosystems).

**CRISPR/Cas9 *EMP3* and *CD44* editing of BEL-A2 cells**. The guide RNA sequences (*EMP3* exon 2—ACTGCTTTTCGTGGCCACTT and exon 5—GATC TATGCCATTCACGCCG) specific to exons 2 and 5 of the *EMP3* gene were each obtained as sense and antisense single oligonucleotides (Eurofins Genomics). Corresponding sense and antisense guide oligonucleotides were annealed by heating at 95 °C for 10 min and allowing the oligonucleotide mixture to cool slowly to room temperature. The annealed guide DNA oligonucleotides were then diluted 1:100 in cold 1× Tris-EDTA buffer (Sigma-Aldrich). The pSpCas9(BB)-2A-puro (PX459) V2.0 vector (Addgene plasmid #62988; kind gift from Zhang Lab) was

linearised using the *Bbs*I restriction enzyme site and the diluted guide DNA oligonucleotides were cloned into the linearised vector using the In-Fusion® HD Cloning Kit (Takara Bio Europe) according to manufacturer's instructions. BEL-A2 cells ($1 \times 10^6$ cells per reaction) were resuspended in supplemented Nucleofactor™ Solution L (Lonza) according to manufacturer's instructions. A total of 10 µg of DNA was added to the cell suspension and gently mixed. The cell/DNA suspension was electroporated using the Amaxa™ Nucleofactor™ II device (Lonza) by applying programme X-001. Transfected cells were transferred back into warm expansion media[13] at 37 °C for 24 h. Drug selection was performed 24 h post-transfection using culture media supplemented with 0.5 µg per ml puromycin for 48 h. Once stable cell growth was observed, transgenic cells were cloned by limiting dilution in concentrations of 0.5 cells per well and 1 cell per well and dispensing the diluted cell suspensions at a volume of 100 µl per well in 96-well plates. The presence of KO transfectant cell population was determined by flow cytometry and confocal microscopy and confirmed by Sanger sequencing of *EMP3*.

For *CD44* gene editing, a single guide RNA sequence (CTACAGCATCTCTCGGACGG) specific to exon 2 of the *CD44* gene was cloned into the pSpCas9(BB)-2A-puro vector as described above. Totally, 10 µg of DNA was transfected into $1 \times 10^6$ BEL-A2 cells as described above, but with the following modifications. The CD34 Cell Nucleofector™ kit (Lonza) and the Amaxa™ Nucleofactor™ II programme U-008 were used for electroporation and puromycin selection was performed for 30 h. Following cloning of the transgenic cell lines as described above, the presence of *CD44* KO clones was confirmed by Sanger sequencing of *CD44* exon 2 and flow cytometry.

**Overexpression of EMP3 in the Daudi cell line**. The full coding sequences of wild-type *EMP3* and an *EMP3* variant in which a premature stop codon had been introduced (c.123 C > G; identified in four MAM negative individuals) were synthesised and cloned into the GFP-positive pEF1α-IRES-ZsGreen1 vector (Clontech) after *EcoR*I and *BamH*I digestion[11]. TOP10 chemically competent bacteria (Invitrogen) were used for transformation and positive colonies were verified by digestion and Sanger sequenced to confirm the insert. pEF1α-IRES-ZsGreen1 empty vector (mock), pEF1α-IRES-ZsGreen1 EMP3 WT or pEF1α-IRES-ZsGreen1 EMP3 mutant containing the premature stop codon (c.123 C > G) were prepared using the plasmid maxi kit (Qiagen). Daudi cells (kind gift from Prof. Urban Gullberg, Lund University) were cultured in RPMI 1640 (Gibco, Life Technologies) supplemented with 10% foetal bovine serum (FBS; Gibco) at 37 °C and 5% CO2. A total of $3 \times 10^6$ Daudi cells were mixed with 10 µg of pEF1a vector, electroporated using the Gene Pulser II electroporation system (Bio-Rad) and incubated for 2 days in one well in a 6-well plate (Nunc). On day 3, the transfected cells were cultured in the presence of geneticin, G418, (InvivoGen) at 400 µg per ml for 3–5 days to achieve positive selection for the transfected cells. On day 5, GFP-positive cells were sorted by a fluorescence-activated cell sorter and cultured in the presence of G418 to assure selection for vector-positive cells. A suspension of $1–2 \times 10^5$ Daudi cells in PBS with 1% bovine serum albumin (BSA) was tested with an anti-MAM eluate, or normal human IgG (as isotype control; R&D systems). Allophycocyanin (APC)-conjugated AffiniPure F(ab')2 fragment goat-anti-Human IgG (Jackson ImmunoResearch Europe) was used as secondary antibody. Analysis of 50,000 events was performed with the LSR Fortessa II cell cytometer (Becton Dickinson) using FacsDIVA software (Becton Dickinson) and analysed with FlowJo Version 10.4.2 (BD Biosciences).

**CD34+ cell isolation and erythroid cell culture protocols**. At IBGRL, CD34+ haematopoietic progenitor cells for three-stage erythroid cell culture were obtained from two MAM-negative whole blood samples (48 ml P9 and 20 ml P10, each with two age- and gender-matched controls sourced at the same time). In addition, whole blood units (P9) or buffy coat (P10) were obtained and transported together with their respective matched controls. CD34+ cells were isolated by positive selection with the MiniMACS magnetic bead system (Miltenyi Biotec). Culturing protocol followed[18], with Biochrom Iscoves modified Dulbecco's medium (IMDM; Source BioScience) including 3% (v/v) AB serum (Sigma), 2 mg per ml human serum albumin (Irvine Scientific), 10 µg per ml insulin (Sigma-Aldrich), 3 U per ml heparin (Sigma-Aldrich), 200 µg per ml transferrin (R&D Systems). In the first stage (days 0–11), the medium was supplemented with 10 µg per ml recombinant human (rH) stem cell factor (SCF; Sigma-Aldrich), 1 µg per ml rH interleukin 3 (R&D Systems), 3 IU per ml erythropoietin (EPO; Roche Products). For the second stage (days 11–14), 10 µg per ml SCF and 3 IU per ml EPO was further supplemented with additional 800 µg per ml iron saturated transferrin, and in the final stage (days 14–21), with 3 IU per ml EPO and additional 800 µg per ml iron saturated transferrin. Initially, $5 \times 10^3$ CD34+ cells were seeded into T25 static tissue culture flasks (BD Falcon Cell Culture Products) and maintained at densities in the range of $0.5–1 \times 10^5$ cells per ml in 5% CO2 at 37 °C. Once the culture volume reached 100 ml the cells were transferred into stirred glass spinner vessels (Integra Biosciences) and maintained at $1–6 \times 10^6$ cells per ml, with passaging carried out as necessary. During the 21-day culture period, cells were counted daily and aliquots were removed for morphological examination (from microscope slides stained with May-Grünwald and Giemsa; Sigma-Aldrich), confocal microscopy and flow cytometric analysis. At the end of cultures, cells were filtered using a standard leucofilter (Pall WBF, Haemonetics Ltd.).

At Lund University, anonymized peripheral blood mononuclear cells (PBMCs) from donated whole blood units prepared in the Reveos automated blood processing system were obtained from the leucocyte waste bag following Lymphoprep (Fresenius Kabi) gradient separation. PBMCs were further enriched for CD34 expression using magnetic beads (Miltenyi Biotec) according to the manufacturer's protocol. Equal numbers of CD34+ cells from each donor were cultured in a three-phase erythroid culture system (modified from Giarratana et al. and Flygare et al.)[31,32] and maintained at a concentration of $1–2 \times 10^5$ cells per ml throughout the culture. The base culture medium was serum-free expansion media (SFEM, STEMCELL Technologies). In phase I (days 0–7), SFEM was supplemented with 50 ng per ml SCF, 50 ng per ml thrombopoietin (TPO), 50 ng per ml FLT3-Ligand (FLT3L), 5 ng per ml interleukin 3 (IL-3) (all purchased from Peprotech) and 100 nmol per l dexamethasone (Sigma-Aldrich). In phase II (days 7–14), SFEM was supplemented with 50 ng per ml SCF, 2 IU per ml EPO (Johnson-Johnson) and 100 nmol per l dexamethasone and in phase III (days 14–18) with 30% FBS (Hyclone, Fisher Scientific UK Ltd.), 3 IU per ml EPO and 300 μg per ml holo-transferrin (Sigma-Aldrich).

**Immunoprecipitation and immunoblotting.** RBC membranes, sodium dodecyl sulfate polyacrylamide gel electrophoresis, immunoprecipitation and immunoblotting were as described in Spring and Reid[33], except that mini gels and MagicMark™ XP Western Protein Standard (ThermoFisher Scientific) were used throughout. The blocking buffer for human eluates and the relevant secondary antibody contained 5% (w/v) γ-globulin-free BSA fraction V (BSA-block), and the protein content of membrane preparations was determined using a commercial protein assay kit (Pierce™ BCA, Pierce Biotechnology) according to the manufacturer's instructions. Primary antibodies were BRIC222 and KZ-1 (both anti-CD44, IBGRL Research Products) and AC-15 (anti-β-actin, Abcam). Anti-MAM eluates were prepared using Gamma ELU-KIT® II following the manufacturer's instructions and diluted with BSA-block. Blots were reprobed with anti-β-actin after prior incubation and imaging with either anti-CD44 or anti-MAM. Secondary antibodies were alkaline phosphatase-conjugated F(ab')₂ fragments of goat anti-sera to mouse IgG1 and human IgG (Jackson ImmunoResearch Europe). Membranes were developed with Western Lightning CDP Star Chemiluminescent Reagent (Perkin Elmer) and images were recorded on a Kodak imager using Kodak imaging software. Proteomic analysis of immune precipitates prepared from normal RBC using anti-CD44 (BRIC222) and an isotype control (IgG1; Sigma-Aldrich) was performed at the University of Bristol Proteomics Facility by Dr. Kate Heesom.

**Platelet preparation and activation.** Anonymised platelet samples from MAM-negative (P10) and MAM-positive donors were obtained from buffy coat platelet concentrates prepared for clinical use at the Dept. of Clinical Immunology and Transfusion Medicine in Lund, Sweden, following approval by the blood centre. Platelet suspensions (10 μl aliquots) were obtained by a sterile procedure from the tube segments and diluted 1:100 in PBS and counted in a haematology analyser (Sysmex kH3; Sysmex Europe GmbH). Equal number of platelets from each sample was centrifuged at $800 \times g$ and washed twice with PBS. Each sample was incubated with 2 mM thrombin receptor activating peptide (TRAP; S8701-Sigma-Aldrich) or PBS for 20 min at room temperature, then stained with human anti-MAM eluate (1:4), human AB serum (1:4) or human IgG (0.2 μg per ml) for 30 min at room temperature. The platelets were washed with PBS containing 1% BSA and stained with mouse anti-human CD62P (P-selectin, clone AK4; BioLegend) (1:100) and goat anti-human secondary conjugated to APC (Jackson ImmunoResearch Europe) (1:200) for 20 min at room temperature. Platelets were washed with PBS and analysed by flow cytometer FACSCanto II (Becton Dickinson).

**Confocal microscopy.** All washes and dilutions were performed in Buffer A (PBS pH7.4 containing 5 mg per ml BSA and 1 mg per ml glucose). Cells were seeded on 0.01% (w/v) poly-L-lysine (Sigma-Aldrich) coated coverslips and incubated for 30 min at 37 °C in 5% CO₂. Cells were fixed in 1% formaldehyde (TAAB Laboratories Equipment) and permeabilised in 0.05% saponin (Sigma-Aldrich). After permeabilisation, all subsequent washes and antibody dilutions were carried out in Buffer A containing 0.005% saponin. Secondary antibodies used were goat anti-mouse Alexa Fluor® 488 and goat anti-rabbit Alexa Fluor® 546 conjugated antibodies (Invitrogen) diluted in 4% (w/v) normal goat serum. Coverslips were mounted on Vectashield® Mounting Medium (Vector Laboratories). Samples were imaged at 22 °C using 40x oil immersion lenses (magnification 101.97 μm at zoom 3.8, 1.25 NA) on a Leica DMI 6000 inverted microscope with phase contrast connected to a Leica TCS SP5 confocal imaging system (Leica). Control slides were used to set the fluorescent levels and these were kept to subsequently image the patient samples. Images were obtained using Leica LAS AF software and subsequently processed using Adobe Photoshop (Adobe) and Volocity v.6.3 (Perkin Elmer).

**EMP3 homology modelling.** The initial homology model was built using SWISS-MODEL automated model building based on claudin-19 coordinates (PDB ID 3×29)[34]. Following inspection of the obtained model and in particular with regard to sequence identity within the first extracellular loop (residues 25–65), this first extracellular region was remodelled from the initial homology model using MODELLER 9.19 and subsequently embedded in a representative cytoplasmic membrane composed of (mole percentages) 50% phosphatidylcholine/25%

cholesterol/15% phosphatidylethanolamine/10% phosphatidylserine (inner leaflet only) using CHARMM membrane builder.

Individual steps in the model building and membrane assembly are listed and referenced in Supplementary Table 13.

**Molecular dynamics calculations and analysis.** Molecular dynamics calculations were performed at 310 K and constant pressure of 1 atm as NPT ensembles using NAMD 2.11 and employing CHARMM36 force field parameters. The system was solvated using a TIP3 water model with a minimum of 10 Å padding in z-axis (membrane normal), neutralised for charges and the ion concentration adjusted to 150 mM NaCl using VMD. The initial assembly was minimised by a six-step equilibration procedure as implemented in the CHARMM membrane builder. Bond lengths involving hydrogen atoms were constrained using SHAKE algorithm as implemented in NAMD. Periodic boundary conditions were employed. The initial structure was energy minimised by applying 1000 steps of conjugate gradient minimisation. For equilibration, electrostatic potentials evaluated every 1 fs using a particle mesh Ewald summation (grid spacing 1.0 Å), this was extended to every 4 fs for production runs. Short-range non-bonded interactions were evaluated every 1 fs during equilibration and every 2 fs during production runs. A cut-off for exclusion of long-range interactions of 12 Å and 10 Å was applied for equilibration and production runs, respectively. A switching function for zeroing of the non-bonded potentials was used at 10 Å (equilibration) and 7.5 Å (production run). Temperature of the system was controlled by Langevin thermostat with a temperature coupling constant of 1. Production run molecular dynamics were calculated for 150 ns simulation time. Diagrams were prepared using VMD (trajectory analysis and interaction with lipids) and Protter (2D representation of topology). The membrane thickness was assessed using MEMBPLUGIN as implemented in VMD, using the distance between phosphate atoms on opposite leaflets of the bilayer as the metric for membrane thickness. Individual steps and software packages used are listed and referenced in Supplementary Table 13.

**Reporting summary.** Further information on research design is available in the Nature Research Reporting Summary linked to this article.

## Data availability

Nucleotide sequence data are available in the DDBJ/EMBL/GenBank databases under the accession numbers MN175570, MN175569, MN164487, MN164486 and MN121937. All other data supporting the findings of this study are available from the corresponding author upon reasonable request. Source data are provided with this paper.

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

## Acknowledgements

The authors would like to thank the following individuals: Patient P9 and her family, and other participating blood donors for their interest and willingness to supply samples for this study; Thomas W. Brady (UVA Blood Bank, Virginia, USA) for provision of archived samples; Professor Jan Frayne, Professor David Anstee and Dr. Kongtana Trakarnsanga for provision of BEL-A2 cell lines which they created with funding from the Wellcome Trust (grant numbers 087430/Z/08 and 1026(10), NHS Blood and Transplant and Department of Health England) and for providing Daudi cells; Professor Urban Gullberg (Lund University) for providing Daudi cells; Dr. Olusogo Busari (University of Bristol M.Sc. student) for Western blotting; Dr. Kate Heesom from the University of Bristol Proteomics Facility for proteomic analysis of immune precipitates and for providing BEL-A2 EMP3, CD81 and GYPA proteomics data; IBGRL Molecular Diagnostics Department for Sanger sequencing support; New Zealand blood centre staff for help with organising and sending whole blood donations from P9 and anonymized healthy donors; Staff at the Department of Clinical Immunology and Transfusion Medicine in Lund, Sweden, for help with provision of anonymized waste products following whole blood donations from healthy donors; Dr. Orly Zelig, Transfusion Service & Blood Bank Director, Hadassah Medical Center, Jerusalem, Israel, for her role in the case of P2. This study was partly supported by the National Institute for Health Research Blood and Transfusion Research Unit (NIHR BTRU) in Red Cell Products (IS-BTU-1214-10032). The views expressed are those of the authors and not necessarily those of the National Health Service, NIHR, or the Department of Health and Social Care. This study was partly supported by the Knut and Alice Wallenberg Foundation (2014.0312 to M.L.O.), the Swedish Research Council (2014-71X-14251 to M.L.O.) and governmental ALF grants (ALFSKANE-446521 to M.L.O.) to the university healthcare in Region Skåne, Sweden.

## Author contributions

N.T., V.K.C., L.T. and C.A.G. conceived, designed and coordinated the study. N.T., B.J., R.L. and C.A.G. performed serology experiments. L.T. performed whole exome sequencing and associated data analysis. L.T., V.K.C., B.K.S. and N.T. performed genetic analyses. V.K.C. and C.A.G. carried out primary and BEL-A2 cell cultures and with B.K.S. performed shRNA experiments. C.L.T. and B.K.S. designed and performed CRISPR/Cas9 experiments. R.E.G. and C.L.T. carried out confocal microscopy. C.A.G., C.L.T., V.K.C. and A.G.A. performed flow cytometry experiments. P.W. carried out protein modelling and molecular dynamics calculations and provided guidance for optimisation of serological testing with monoclonal antibodies. F.S., N.T. and J.R.S. designed and performed Western blotting. F.S. carried out immunoprecipitation experiments. N.T. designed and carried out MAIEA experiments. M.M. provided bioinformatic analysis of *EMP3* alleles. A.G.A. performed overexpression, platelet experiments and supplementary P10 cell culture. J.R.S. provided guidance and reagents for overexpression experiments. C.M.W. and C.L.F. identified and provided MAM-negative samples. M.L.O., L.W., R.C., V.Y., U.F., A.S. and B.M. detected and referred samples for antibody identification. L.W., M.L.O. and R.C. provided additional samples and clinical data. M.L.O. provided guidance for overexpression and supplementary P10 cell culture. D.J.A. and M.L.O. provided advice on experimental design, analysis and interpretation of data, manuscript preparation and critical review. N.T., V.K.C., L.T., C.A.G., J.R.S., M.L.O. and D.J.A. wrote the paper. All authors contributed to review of the final paper.

## Competing interests

A patent application (PCT Application No. PCT/GB2019/053102) covering ex vivo erythroid cell culture proliferation enhancement due to lack of or disruption of EMP3 has been filed (N.T., C.A.G., L.T., V.K.C. and D.J.A.). The remaining authors declare no competing interests.
