## [Peer Review File · Nature Communications]

Reviewers' Comments:

Reviewer #1:

Remarks to the Author:

In the paper by Thornton et. al, the authors evaluate the association of EMP3 in erythroid proliferation as well as in regulation the MAM negative phenotype. The MAM phenotype was originally discovered in the mid 1990's as an antigen that was missing from a small cohort of women who went on to have shorten RBC survival and some association with hemolytic disease of the newborn. In the original study, 3 patients were examined (Transfusion. 2000 Sep;40(9):1132-9). In this new study, the authors examine 10 MAM negative individuals and find that all 10 show that EMP3 may be a candidate to help regulate MAM expression. The premise of the article is novel, but the experiments as presented severely dampen my enthusiasm. Moreover, the paper at times seems to suffer from a lack of direction as the authors claim an association between CD44, EMP3, and the MAM negative phenotype, but they do not follow assessment of all three of these markers throughout the paper. This makes it very difficult to draw any conclusions.

Major criticisms:

1. What are EMP3 levels in BEL-A2 cells? Total protein levels need to be given.
2. The authors claim that EMP3 and CD44 staining overlap in BEL A2 cells (lines 106-107), but the staining in Supplementary Fig. 1 is difficult to assess. A nuclear staining is necessary to properly determine the location of the CD44 and EMP3 staining. Some of the staining in panel C for CD44 looks nuclear while it is difficult to know if the EMP3 staining shown is cell associated. The CD44 staining in panel D also does not appear, as the authors claim, similar between the EMP3 sh4 and scr cells (differences in the number of cell stained).
3. The authors write for supplementary figure 2 that there is a marked reduction in expression in expression of EMP3 on BEL A2 cells. However, I do not believe that 10^0 expression by flow is real expression and am not convinced that these cells even express EMP3. Further, the flow staining in these cells is not consistent between supplementary figure 1 and supplementary figure 2 although no isotype control is shown in Supplementary Figure 1.
4. In addition, no staining is detectable on my screen in Supplementary Figure 2B in any of the cells even after zooming in. In Supplementary figure 2 panels C and D, EMP3 staining should be shown to estimate protein expression between endogenous and overexpressing cells.
5. Given the low expression of EMP3 on BEL A2 cells, it would have been interesting to see EMP3 expression in erythroid development in Supplementary Figure 5.

Minor points:

1. The authors write that Supplementary Figure 4 shows a direct association between MAM glycoprotein and CD44 by immunoprecipitation with subsequent immunoblotting (lines 187-188). However, the figure legend of Supplementary Figure 4 only states that an immunoblot for CD44 expression in 9 total patients was done. In addition, the single band for CD44 is interesting given the heterogenous glycosylation and splicing patterns that have been shown for this protein (<https://doi.org/10.1186/s12014-018-9198-9>). What size band did the authors observe? And, what was the relative level of EMP3 on these cells?
2. The introduction and conclusions as written are hard to follow. Do the authors believe that EMP3 is more similar to PMP22/clauidins (where it would be found perhaps in junctions between cells) or perhaps a tetraspanin (as eluded to in the discussion)? It would have been helpful to choose a direction as the authors discuss both sets of four transmembrane protein families but it is unclear which is more similar to EMP3. Figure 5 is the most compelling data of the paper showing an association of CD81 and CD44. This would have been interesting to explore as I wonder how much sequence homology exists between CD81 and EMP3.
3. The authors should expand on potential "factors external to the developing red cells" which may explain why MAM negative individuals do not show any apparent erythrocytosis or reticulocytosis (lines 180-183) but show differences in proliferation when put in culture (Figure 4).
4. Figure 1 should be included as Supplementary data and a figure similar to panel 1A has already been published (PMCID: PMC5362274)

Reviewer #2:

Remarks to the Author:

The paper by Thorton et al., reports on the identification of mutations in the epithelial membrane protein 3 (EMP3) as a putative genetic basis for the MAM-negative red blood cells' phenotype. MAM-negative individuals are rare and manifest no apparent disease. However, it is highly relevant in pregnancy. Since the Indian blood group system CD44 seemed weakly expressed in MAM-negative red cells, a link between EMP3 and CD44 was suggested.

The authors observe a decrease in CD44 expression during the maturation of erythroblasts. The decrease in CD44 expression seems to be linked to the enucleation. But in the case of MAM-negative individuals, this decrease might also be explained by the accumulation of CD44 in vesicles. The authors suggest a possible influence of EMP3 on the stability of CD44.

Even though the authors show a clear difference in the distribution of CD44 during erythroblast division (Fig5), a statistical analysis of these observations would be desirable.

From the supplementary figure 4, it does seem that the expression of CD44 is decreased in the MAM-negative samples. However, a claim of a direct association between the MAM glycoprotein and CD44 seems to be an overinterpretation of the data. The experiment described in Supplementary figure 4 is not clear and the sentence "Immunoprecipitation with subsequent immunoblotting provided evidence of a between the MAM glycoprotein and CD44 in normal MAM-positive red cells (Supplementary Figure 4) " is misleading. The lysates of MAM-negative individuals seem to have been blotted with a CD44 antibody. No co-immunoprecipitation or any other protein-protein interaction studies are shown. In addition, in the Mat&Met, a membrane preparation seems to have been prepared. On which of these fractions was the beta-actin staining done?

Several other questions remain. All mechanistic relationships between CD44 and EMP3 are highly speculative. In order to support their hypothesis, several experiments are lacking. How is this potential influence of EMP3 on CD44 stability exerted? qPCR shows that this is not a transcriptional effect. Are the CD44 containing vesicles recycled or targeted for degradation.

What is the consequence of the decrease in CD44 expression. Loss of function experiments focused on CD44 would be very helpful there.

Altogether, the authors report on an interesting observation. However, the results are very preliminary and cannot be published at this stage.

Reviewer #3:

Remarks to the Author:

The authors demonstrate that the EMP3 protein carries the MAM bloodgroup antigen. They provide the structure of EMP3, the function in erythroblast proliferation and differentiation, and the interaction with CD44. The data are placed in the context of erythropoiesis but also in a general context given the association of EMP3 expression and cancer progression.

The paper is well written, the data are convincing.

The coherence of the data is the central molecule. The paper is a bit a collection of tiny stories that make up "all you want to know on EMP3 ". It is, however, important to know the basis of this novel bloodgroup antigen and therefore all aspects of EMP3 are important.

In figure 1 the authors show the structure of EMP3, and in figure 2 the variations in MAM-negative individuals. To me it would be more logic to chance the order of these figures. Once we know which genetic variations are present, we can look at the structure, and the variations that do not lead to a null variant. I am a bit confused at this point. The text mentions the null alleles, and variations in amino acids (such as Tyr41Ter). Only when I inspected the supplemental tables for a second time, I realized that the MAM-negative individuals are null (although .. what about the I125V variant?), and other variants are found in genome dataset without a link to MAM-antigen

expression. This should be made more explicit in the manuscript text .

The authors also show that loss of EMP3 enhances the proliferation of erythroid cells. These data are shown in figure 4. I have some remarks to this figure.

> Proliferation, cell division, is a logarithmic process. The linear scale of figure 4A exaggerates the final difference, but fails to show what happens at the start of the culture. To my opinion, growth curves must always be displayed on a logarithmic scale to be able to evaluate the expansion throughout the culture period.

> Figure 4B-left panels give a nice demonstration of 4A. The right hand panel makes the point that lack of EMP3 yields many more reticulocytes after leukofiltration. BUT these data are not quantitative, and it is unclear what causes this excess of reticulocytes. How many cells were harvested before filtration and what was their enucleation rate? And how many enucleated cells were recovered after filtration.

Rather than showing a pellet of reticulocytes, the authors should show how an equal amount of erythroblasts cultured for e.g. 12 days then differentiates under explicit differentiation conditions

> Figure 4C shows that cells with or without EMP3 have similar differentiation profiles, and the authors suggest that it is not a delay of differentiation that causes an excess of erythroid cells after a 20 day culture. The difference in yield, according to figure 4A is 3 to 5-fold and because of the linear scale we can not interpret from where this difference originates. The authors should include a viability screen (flow cytometry?) and/or a cell division assay (using for instance CFSE) to get an idea why cell yield is different.

> The materials and methods show two in vitro culture protocols. The legend says that 4A is the three stage culture; and the materials and methods say that phaseII was changed to phase III on day 14. This fits with the sudden halt of proliferation on day 14.

I cannot believe, however, that the morphological analysis of 4C is from the same culture. It is surprising that there are already enucleated reticulocytes in presence of dexamethasone which strongly impairs enucleation. It seems that the morphological analysis of 4C stems from the Bristol type of culture which should be stated.

Finally, the authors examine the interaction with CD44. At this stage it would be more clear when the authors would state that they use EMP3-negative cells. The cells may not only have lost the MAM antigen, I suppose they lack the entire protein (after all, missense mutations were found as well)

Figure 5 is based on single cells. The authors may want to show additional data in the supplemental figures and/or be a bit more modest in their interpretation

In the discussion the authors try to interpret why CD44/EMP3 expression regulate erythropoiesis in vitro but not in vivo and comment on the lack of extracellular matrix in a culture flask. However, in vivo Epo levels drop when oxygen saturation is sufficient. Because lack of EMP3 is unlikely to render cells Epo independent, the Epo concentration will override the importance of EMP3 in vivo

Reviewer #4:

Remarks to the Author:

Thornton et al. in an overall well written manuscript provide strong genetic evidence that EMP3 is the gene responsible for expression of the previously unsolved MAM blood group, further supported by in vitro experimental evidence that the expressed EMP3 gene product is the target of anti-MAM antibodies.

The significance of this work is that only a handful of blood group antigens remain molecularly unsolved, of which MAM is one. MAM-negative phenotypes are very rare (i.e. it is a high incidence

antigen). 10 known MAM-negative individuals were studied, which substantially represents the collective MAM-negative experience, using whole exome sequencing; 5 different predicted EMP3 null-variants (1 gain-of-stop, 2 partial gene deletions, and 2 different full gene deletions) were identified. Every MAM-negative subject tested was homozygous for an EMP3 LOF variant, and nonsense variation in this gene in gnomAD is plausibly rare.

Thornton et al. then make a compelling case using multiple lines of evidence that EMP3 expression confers MAM antigen(s). They show that a monoclonal anti-EMP3 fails to react with MAM-negative patient RBCs but does react with reference RBCs, and that anti-MAM patient antibodies block the binding of anti-EMP3 mAb to normal RBCs. Compellingly, the authors are using anti-MAM containing eluates from the allosensitized MAM-negative patients – this is a real strength in the experimental design. The authors further show that shRNA EMP3 knock-down reduces MAM expression in the immortalized erythroid progenitor cell line BEL-A2, and that CRISPR/Cas9 disruption of the EMP3 gene in Daudi cells (a Burkitt cell line) abolished MAM expression, and that expression of EMP3 cDNA in these cells rescues anti-MAM reactivity.

The authors also made observations around role of EMP3 in erythropoiesis in cultured primary erythroid progenitors from 2 patients and 4 normal subjects find increased ex vivo proliferation. Pursuing the possible association with CD44, in normal RBCs they find that EMP3 and CD44 co-IP and that loss of EMP3 expression results in aberrant CD44 localization (diffuse cell surface staining and new co-localization with CD81 in vesicles) and apparent differences in CD44 abundance with differentiation (up in early stage cells, more markedly down in more mature erythroid cells). The authors speculate that EMP3 may be needed to stabilize or appropriately localize CD44 in the membrane, and therefore loss of EMP3 would lead to lower CD44 expression in mature cells. Erythroid progenitor cultures from MAM-negative subjects further demonstrated increased proliferation in vitro – which follows what was previously thought to be the function of EMP3 in oncogenesis, although there is no evidence of erythrocytosis, reticulocytosis, or increased risk for malignancy in the MAM-negative individuals so the in vivo significance of this finding is unclear. Taken together this is a compelling paper that MAM blood group is encoded by EMP3 using multiple lines of evidence.

Comments:

- The recruitment of materials from 10 MAM-negative subjects is a testament to a huge collaborative effort and the authors are to be commended
- Throughout, please use HGVS nomenclature to describe the genetic variants and provide the IDs for the genome and transcript references used
 - o Please avoid the use of the word “mutation”, which is problematic because of entanglements with frequency and/or function in common definitions, suggest the use of variant and its permutations
- Family information:
 - o Please indicate subject relatedness either by coding family IDs into each unique subject identifier or by using a separate family/pedigree identifying number
 - o Please provide the relationship to the proband in a separate column in the supplemental tables
 - o Please define P and F subjects in the first supplemental table
- All subjects are homozygous for their null EMP3 variant, which is a little surprising given the rarity – this suggests either consanguinity or that there could be heterozygosity for deletions (hemizygous)
 - o Please verify there is no evidence for loss of read depth in the WES and also note if there is heterozygosity of other regional variants
 - o Please also comment on presence or absence of consanguinity (if not known please state) – if unknown relatedness could be assessed using the WES if this is very clearly OK with the

consent/IRB and the study subjects to do such an analysis

o Alternately, acknowledging some limitations of WES, a haplotype analysis could lend insights into if similar variants have arisen independently or if they appear to share a common founder allele

- On page 4, the exclusion of CD44H coding variants in WES has no background for the reader (presumably this was tested because Indian is weak and then also because of the later CD44 findings).

o CD44H is an isoform of CD44 and not a discrete gene, please indicate what WES findings were for the whole gene – it's not been determined that the CD44 part of this EMP3 story is actually restricted to CD44H

o This could be in the supplement along with any other candidate gene findings that were filtered in identifying EMP3

- I am glad to see the gnomAD frequencies

o Please correct the numbers to accurately reflect what is currently reported in gnomAD – for example, there are 43 p.Tyr41Ter alleles out of 251,000 subjects, not 44 out of 245,818 – this is of no consequence to the authors' conclusions, but it should be corrected before publication

o the authors should mention in the supplement that the annotation in gnomAD only has 4 exons while the reference transcript used elsewhere has 5 exons (I'm assuming the difference is the UTR), this is avoid confusion to readers (harmonizing with HGVS will also help avoid confusion).

o The authors could also note that there were no homozygous LOFs in gnomAD

o Although it is true that four subjects with EMP3 c.123C>G were detected these were not independently discovered and not related to population frequency as at least some are related, this should be considered when discussing frequency on page 5

- The authors describe MAM in a way that implies it as being generally expressed on hematopoietic cells, but the findings in Montgomery et al. are not consistent with MAM being expressed on platelets. Please correct this.

- Figures – please increase font sizes for readability

o Figure 1 is probably useful for others that will study EMP3 in the future but I don't know if it adds much to findings here, so if there are needs in revision it could be kept but could go in the supplement. If this is kept please add the predicted N-linked glycosylation sites

o Figure 2

♣ please increase or add more arrows to indicate gene direction in both panels

♣ for the upper box see above comment about coverage depth and possible missed compound heterozygous deletions above – for instance, in P9 the upstream gene might not have good coverage relative to the control, but downstream (from what can be seen) looks somewhat similar

♣ please indicate in the legend that light blue indicates UTRs

o Figure 3 please indicate which anti-MAM (an eluate I think) is being used, and in 3b please give AA translation described in the text in the figure

o Figure 4 there should be more clarity on the experiment that resulted in the cell pellet picture (is this after culture of a normalized seeded plate?)

o Figure 5b it looks like there is a big difference in CD81 staining in MAM-negative cells but this is not discussed by the authors ... I mention this because it seems noteworthy as another change of loss of EMP3, if this is representative, and CD81 also has a proliferative story

♣ In speculating on mechanism, another role for EMP3 may be appropriate intracellular trafficking, particularly given the intracellular CD81/CD44 findings

o The experiments in supplemental figure 2 are compelling and I think should be in the body of the main paper – my summary of why is that this is the CRISPR/Cas9 KO of EMP3 in Daudi cells with loss of EMP3 mAb staining showing it is KO, then overexpression by transfection of EMP3 and gain of anti-MAM seroreactivity with a good EMP3 nonsense control – this is the clincher experiment they have the right gene – knock out then rescue using a very good reagent (the human eluate with anti-MAM)

o Supplemental figure 4

- ♣ please show the IN weak results
- ♣ in the main text and methods this figure is referred to as an IP experiment with EMP3 and CD44 but in the figure itself looks like a regular blot – please revise to indicate exactly what was done and clarify the conclusions

- Supplemental table 4 and PNGase F

o Page 15 describes RBC treatment with papain and PNGase F, and the results are seemingly in Supplemental table 4, but this is a confusing table – is everything in this result treated with papain/PNGase F?

- ♣ Please clearly show the results of the treated and untreated cells.

- ♣ I am very interested in clarifying the PNGase F results because treatment with PNGase F has been reported (in AABB meeting abstract form) to result in ablation of anti-MAM activity on multiple cells and collapse of the smeary bands on anti-MAM immunoblots to discrete bands – I think from this I would have expected some loss of anti-MAM eluate reactivity with PNGaseF. Although there are limitations in that the AABB abstract findings it is a rare piece of data associated with this blood group,so for the purposes of examining this as a blood group it is important to clarify what are the PNGase F findings in this context

- Supplemental table 5 please also include the detection of MAM on cells of different lineages from the MAM blood group publications in addition to cancer lineages, which given that malignancy does not seem to be part of the MAM blood group or erythroid story right now and normal tissue expression patterns are of interest

- If the authors need room in revision, they could summarize the PMP22 story more succinctly.

Author response for manuscript NCOMMS-18-37307A-Z

Reviewer comment:	Author response:
Reviewer #1 - expert in EMPs (Remarks to the Author): In the paper by Thornton et. al, the authors evaluate the association of EMP3 in erythroid proliferation as well as in regulation the MAM negative phenotype. The MAM phenotype was originally discovered in the mid 1990's as an antigen that was missing from a small cohort of women who went on to have shorten RBC survival and some association with hemolytic disease of the newborn. In the original study, 3 patients were examined (Transfusion. 2000 Sep;40(9):1132-9). In this new study, the authors examine 10 MAM negative individuals and find that all 10 show that EMP3 may be a candidate to help regulate MAM expression. The premise of the article is novel, but the experiments as presented severely dampen my enthusiasm. Moreover, the paper at times seems to suffer from a lack of direction as the authors claim an association between CD44, EMP3, and the MAM negative phenotype, but they do not follow assessment of all three of these markers throughout the paper. This makes it very difficult to draw any conclusions.	We thank the reviewer for these comments. We have made substantial changes to the text throughout the manuscript to help explain the links between MAM antigen, EMP3 and their association with CD44. For this purpose, we also included additional data; for example, immunoprecipitation and proteomics data showing direct association between MAM/EMP3 and CD44, MAM expression levels during cell cultures in parallel with CD44 expression, etc. We consider that our data demonstrates clearly that that MAM expression is reliant on intact EMP3 and that anti-MAM and anti-EMP3 recognize the same molecule (as primarily shown by knockdown, knockout and overexpression experiments and antibody inhibition studies confirming that anti-MAM blocks the binding of EMP3 with anti-EMP3). The additional data we provide helps elaborate the CD44 association. The reviewer correctly pointed out that all three markers were not assessed in every experiment. However, we consider that we have provided proof that MAM and EMP3 are in fact the same marker, so in some experiments we have used anti-MAM as a surrogate for anti-EMP3 (due to the poor performance of this monoclonal antibody). We hope the changes that we have made will help better explain the sequence of experiments, relationship between the markers and the conclusions we have drawn.
Major criticisms: 1. What are EMP3 levels in BEL-A2 cells? Total protein levels need to be given.	Measuring the total protein levels was not within the scope of this project, but we were able to use our limited proteomic analysis of BEL-A2 cells to show that EMP3 is present in low abundance in undifferentiated cells, similar to tetraspanin CD81. The ratio of EMP3/CD81 is 1.33 (n= 3 BEL-A2 cell cultures). Gautier et al. (Cell Rep. 16(5):1470-1484, 2016) reported the absolute copy number of 30,000 molecules of CD81 per cell early in erythropoiesis, declining to 3,000 molecules per cell at later stages. Since undifferentiated BEL-A2 cells are predominately pro- to early basophilic erythroblasts, we could infer there are approximately 39,900 EMP3 molecules per cell. Furthermore, the ratio between EMP3 and the standard erythroid marker glycoprotein glycoporphin A (GPA) was 1.38. In BEL-A2 cells, GPA abundance is relatively low and similar to EMP3, but unlike EMP3, it would increase

	if cells were to differentiate. We have added to the text a comment about relative EMP3 levels on BEL-A2 cells. Page 5: 'The undifferentiated wild-type BEL-A2 cells are predominantly in the erythroblast stage of development and express EMP3 at a low level, similar to glycoporphin A and tetraspanin CD81 levels at the same stage'. Proteomics data added as new Supplementary Table 5.
2. The authors claim that EMP3 and CD44 staining overlap in BEL A2 cells (lines 106-107), but the staining in Supplementary Fig. 1 is difficult to assess. A nuclear staining is necessary to properly determine the location of the CD44 and EMP3 staining. Some of the staining in panel C for CD44 looks nuclear while it is difficult to know if the EMP3 staining shown is cell associated. The CD44 staining in panel D also does not appear, as the authors claim, similar between the EMP3 sh4 and scr cells (differences in the number of cell stained).	As suggested by the reviewer, Supplementary Figure 1, now Supplementary Figure 2, has been amended to include a legend describing nuclear staining and using a better resolution for images in panel c. The legend for panel c was changed to: 'Confocal microscope immunofluorescence analysis (magnification 40x) showed no difference in CD44 expression between EMP3 sh4 and scr cells when stained with anti-CD44 on day 5 of differentiation. However, when sh4 and scr cells were stained with anti-MAM on day 5 of differentiation, reduction of EMP3 in sh4 cells was observed.' The reviewer is correct in noticing that there is a different number of cells in CD44/DAPI sh4 and scr panels, but the improved resolution should help the viewer to see the similarity in the intensity of the staining more clearly.
3. The authors write for supplementary figure 2 that there is a marked reduction in expression in expression of EMP3 on BEL A2 cells. However, I do not believe that 10⁰ expression by flow is real expression and am not convinced that these cells even express EMP3. Further, the flow staining in these cells is not consistent between supplementary figure 1 and supplementary figure 2 although no isotype control is shown in Supplementary Figure 1.	The reviewer has observed that Supplementary Figures 1 and 2 differ (now Suppl. Figs 3 and 4). This is correct. The difference is expected because they depict two individual experiments where cells were stained with different antibodies (Suppl. Fig. 3 with anti-MAM and Suppl. Fig. 4 with anti-EMP3). We have established by proteomics that the native BEL-A2 cells express EMP3. KO cells in Supplementary Figure 2 (now 4) indeed do not express any EMP3 as shown by the flow cytometry, confocal analysis and Sanger sequencing of EMP3 in Figure 3 (main paper). In that figure, our main experiment of knocking out EMP3 and staining the resulting KO cells with anti-MAM is shown. As part of due diligence we have stained the same cells with commercially available anti-EMP3 MAb in order to show we tested the available MAbs. Results are shown in Supplementary Fig.2 (now 4) and are consistent with the poor performance we encountered with all commercial MAbs tested (4 different clones), which reflected in the low values seen by flow cytometry. To that effect, we have changed the figure legend to

	remove the term 'marked' reduction and highlighted the problems we encountered using commercial MAbs. Thank you for pointing out the lack of isotype control in Supplementary Figure 1 (now 3), which has now been included.
4. In addition, no staining is detectable on my screen in Supplementary Figure 2B in any of the cells even after zooming in. In Supplementary figure 2 panels C and D, EMP3 staining should be shown to estimate protein expression between endogenous and overexpressing cells.	The staining in images in Supplementary Figure 2b (now 4c) was not clearly visible, especially when in combination with bright DAPI stain. We have adjusted the brightness of the images, applying the same parameters to all images. We hope the staining is now visible. Supplementary Figures 2c and 2d have now moved to Figure 4 (main paper). In Figure 4c the histogram shows MAM reactivity following transfection of Daudi cells with either mock which correspond to the endogenous expression, wild type (WT) or EMP3 mutant. All were compared to unstained cells and to all conditions.
5. Given the low expression of EMP3 on BEL A2 cells, it would have been interesting to see EMP3 expression in erythroid development in Supplementary Figure 5.	We have included data (Supplementary Figure 8) from an adult erythroid culture (P9 and matched control) following the expression of MAM (EMP3) as stained by anti-MAM from day 6 to 18, including reticulocyte cells filtered on day 21.
Minor points: 1. The authors write that Supplementary Figure 4 shows a direct association between MAM glycoprotein and CD44 by immunoprecipitation with subsequent immunoblotting (lines 187-188). However, the figure legend of Supplementary Figure 4 only states that an immunoblot for CD44 expression in 9 total patients was done. In addition, the single band for CD44 is interesting given the heterogenous glycosylation and splicing patterns that have been shown for this protein (https://doi.org/10.1186/s12014-018-9198-9). What size band did the authors observe? And, what was the relative level of EMP3 on these cells?	We thank the reviewer for making this point. The immunoprecipitation data was omitted in error and is now included in Supplementary Figure 7. Figure legend was amended to reflect that blotting was done in six patients (P2, P4, P5, P8, P9, P10) and three random controls (C1-C3). We did observe a single CD44 band on immunoblots of red cell membranes. Red cells express only one CD44 isoform common to all haemopoietic cells, CD44H (CD44s; Leucocyte Antigen Factsbook, Barclay et al, Academic Press, 1997, 2nd edition). Red cell CD44 comprises a major band of approximately 80,000 kDa (77,400-80,750 kDa) and a minor band of 68,500 kDa. These are referred as major 80,000 and minor 70,000 components (Spring et al, Immunology 64(1), 37-43, 1988), sometimes reported as a single band of 80,000 kDa (Telen et al, J Biol Chem 271(12), 7147-53, 1996). The above findings were observed on large gels, while commonly used mini gels do not resolve such fine detail, so two bands of CD44 can appear as a single band, as seen with our results.

	Blotting with anti-EMP3 MAbs was poor, which was consistent with our experience of anti-EMP3 MAbs in flow cytometry, serology, etc. The relative level of EMP3 on these cells can be approximated to the cultured erythroid cells obtained from adult CD34+ progenitor cells and tested by proteomics on day 8 of culture. In that case, the relative level of EMP3 was slightly lower than CD81 (EMP3/CD81 = 0.8).
2. The introduction and conclusions as written are hard to follow. Do the authors believe that EMP3 is more similar to PMP22/clauidins (where it would be found perhaps in junctions between cells) or perhaps a tetraspanin (as eluded to in the discussion)? It would have been helpful to choose a direction as the authors discuss both sets of four transmembrane protein families but it is unclear which is more similar to EMP3. Figure 5 is the most compelling data of the paper showing an association of CD81 and CD44. This would have been interesting to explore as I wonder how much sequence homology exists between CD81 and EMP3.	We have expanded the text to improve the flow of the narrative and ease the clarity of reading. We agree that our description of EMP3, PMP22 and CD81 was confusing and we amended the text on page 5, line 132 and page 7, line 226 to improve this. Any relation between EMP3 and the tetraspanin family of proteins (including CD81) could be decisively dismissed based on the lack of conserved motifs in EMP3. We have briefly explored the homology between EMP3 and CD81. While the two proteins do topologically share some resemblance (with regard to trans-membrane topology), their sequence is disparate as EMP3 lacks the critical cysteine-motifs in the extracellular domain of tetraspanins.
3. The authors should expand on potential “factors external to the developing red cells” which may explain why MAM negative individuals do not show any apparent erythrocytosis or reticulocytosis (lines 180-183) but show differences in proliferation when put in culture (Figure 4).	We have modified the text accordingly on page 7, lines 194-198.
4. Figure 1 should be included as Supplementary data and a figure similar to panel 1A has already been published (PMCID: PMC5362274)	We kept Figure 1 as a part of main manuscript (now Figure 2), because we feel it illustrates well the topology, size and closeness to the cell membrane discussed in the text. Panel 1a is similar to PMCID: PMC5362274 because it was modelled using the same software (PROTTER). We have acknowledged this paper accordingly.

Reviewer #2 - expert in CD44 (Remarks to the Author): The paper by Thorton et al., reports on the identification of mutations in the epithelial membrane protein 3 (EMP3) as a putative genetic basis for the MAM-negative red blood cells' phenotype. MAM-negative individuals are rare and manifest no apparent disease. However, it is highly relevant in pregnancy. Since the Indian blood group system CD44 seemed weakly expressed in MAM-negative red cells, a link between EMP3 and CD44 was suggested.	No response required.
The authors observe a decrease in CD44 expression during the maturation of erythroblasts. The decrease in CD44 expression seems to be linked to the enucleation. But in the case of MAM-negative individuals, this decrease might also be explained by the accumulation of CD44 in vesicles. The authors suggest a possible influence of EMP3 on the stability of CD44.	No response required.
Even though the authors show a clear difference in the distribution of CD44 during erythroblast division (Fig5), a statistical analysis of these observations would be desirable.	We have quantified the amount of fluorescence (CD44 staining intensity) in the cleavage furrow of three MAM-negative (P9) and three control dividing cells obtained with Leica LAS AF software and included the graph and statistical analysis in Figure 6 (previously Figure 5).
From the supplementary figure 4, it does seem that the expression of CD44 is decreased in the MAM-negative samples. However, a claim of a direct association between the MAM glycoprotein and CD44 seems to be an overinterpretation of the data.	We thank the reviewer for pointing this out; the blot of immunoprecipitation was omitted in error and is now included in Supplementary Figure 7. The co-IP data backed by additional proteomics data (added as a Supplementary Proteomics File 1) should now show no over-interpretation.
The experiment described in Supplementary figure 4 is not clear and the sentence "Immunoprecipitation with subsequent immunoblotting provided evidence of a between the MAM glycoprotein and CD44 in normal MAM-positive red cells (Supplementary Figure 4) " is misleading.	Please see the comment above.
The lysates of MAM-negative individuals seem to have been blotted with a CD44 antibody. No co-immunoprecipitation or any other protein-protein interaction studies are shown. In addition, in the Mat&Met, a membrane preparation seems to have been prepared. On which of these fractions was the beta-actin staining done?	Please see the comment above. The co-IP and proteomics data have now been included and Materials and Methods amended. The solubilized red cell membranes from six MAM-negative patients and three controls were analysed by immunoblotting with anti-CD44. This blot was re-probed with anti-actin. IP samples were prepared from red cells incubated with MAbs,

	from which membranes were prepared, solubilized and immune complexes were extracted with Protein G Sepharose. This is now detailed in the Materials and Methods page 23, lines 716-737 and the legend of new Supplementary Figure 7.
Several other questions remain. All mechanistic relationships between CD44 and EMP3 are highly speculative. In order to support their hypothesis, several experiments are lacking. How is this potential influence of EMP3 on CD44 stability exerted?	We would like to emphasize here that the subject of our manuscript is the identification of EMP3 as the glycoprotein required for the expression of the MAM antigen. We have confirmed this relationship through a series of experiments, but for the completeness of the study we have also explored other observed links, such as the potential influence of EMP3-negative phenotype on CD44. Here we have shown the difference in expression levels of CD44 in absence of EMP3 through immunoblotting, co-IP and confocal microscopy. Further investigation of this relationship is beyond the scope of this project and presently we only suggest possible ways of EMP3 influence on CD44 that are consistent with the data presented. Further studies are needed to explore the CD44 link.
qPCR shows that this is not a transcriptional effect. Are the CD44 containing vesicles recycled or targeted for degradation.	This particular aspect of CD44 pathway is subject to future work.
What is the consequence of the decrease in CD44 expression. Loss of function experiments focused on CD44 would be very helpful there.	The function of CD44 in erythroid cells is still unknown. We have developed a CD44 knockout cell line, which will be explored as a separate project.
Altogether, the authors report on an interesting observation. However, the results are very preliminary and cannot be published at this stage.	We agree with the reviewer that our results on EMP3 relationship with CD44 are preliminary, but we would argue that it was not our intention to concentrate on this interaction. The subject of our manuscript is primarily the identification of the EMP3 and MAM relationship. CD44 interaction will be explored further in future work, but we would like to share our observations with the wider research community as they may inform future work by other groups with a CD44 focus.
Reviewer #3 - expert in erythropoiesis (Remarks to the Author): The authors demonstrate that the EMP3 protein carries the MAM blood group antigen. They provide the structure of EMP3, the function in erythroblast proliferation and differentiation, and the interaction with CD44. The data are placed in the context of erythropoiesis but also in a general context given the association of EMP3 expression and cancer progression.	We would like to thank the reviewer for such a positive assessment of our work.

The paper is well written; the data are convincing. The coherence of the data is the central molecule. The paper is a bit a collection of tiny stories that make up “all you want to know on EMP3”. It is, however, important to know the basis of this novel blood group antigen and therefore all aspects of EMP3 are important.	
In figure 1 the authors show the structure of EMP3, and in figure 2 the variations in MAM-negative individuals. To me it would be more logic to change the order of these figures. Once we know which genetic variations are present, we can look at the structure, and the variations that do not lead to a null variant. I am a bit confused at this point. The text mentions the null alleles, and variations in amino acids (such as Tyr41Ter). Only when I inspected the supplemental tables for a second time, I realized that the MAM-negative individuals are null (although .. what about the I125V variant?), and other variants are found in genome dataset without a link to MAM-antigen expression. This should be made more explicit in the manuscript text.	We have followed the reviewer’s suggestions and changed the text in the manuscript to introduce the genetic findings followed by the structural properties of EMP3 protein (pages 4 and 5). The figures (Figure 1 and 2) have also been changed accordingly. We have amended the text on page 4, lines 118-121 and the notes in Supplementary Table 3 to clarify that all MAM-negative individuals tested had EMP3 null alleles [full gene deletions c.1-3513_*1379del and c.1-3532_*1361del, exon deletions c.182-186_322+418del and c.323-231_*421del, and a premature translation termination c.123C>G (p.Tyr41Ter)]. We have now explained in the notes of Supplementary Table 3 that p.Ile125Val variant was not predicted to be expressed because it was located on an allele downstream from the codon for premature termination of protein translation p.Tyr41Ter. In Supplementary Table 4b, it has been clarified that the impact on MAM antigen expression for alleles with only missense variants is not known.
The authors also show that loss of EMP3 enhances the proliferation of erythroid cells. These data are shown in figure 4. I have some remarks to this figure. > Proliferation, cell division, is a logarithmic process. The linear scale of figure 4A exaggerates the final difference, but fails to show what happens at the start of the culture. To my opinion, growth curves must always be displayed on a logarithmic scale to be able to evaluate the expansion throughout the culture period.	We have changed the cumulative growth rate and total cell number proliferation graphs in Figure 5 and Supplementary Figure 6 (previously Figure 4 and Supplementary Figure 3) to a logarithmic scale as suggested by the reviewer.
> Figure 4B-left panels give a nice demonstration of 4A. The right hand panel makes the point that lack of EMP3 yields many more reticulocytes after	The cells in cultures were counted daily and adjusted to the same cell density as necessary, resulting in different volumes of MAM-negative and control samples.

leukofiltration. BUT these data are not quantitative, and it is unclear what causes this excess of reticulocytes. How many cells were harvested before filtration and what was their enucleation rate? And how many enucleated cells were recovered after filtration.	We have expanded the legend to Figure 5 (previously Figure 4) to address the reviewer's comments. We have now explained how all cells were filtered on day 21 using the same leucofilters and filtration method. On day 21, the total number of cells filtered can be inferred from the proliferation graphs. For example, P10 had 8.7×10^8 cells and C1(P10) 3.2×10^8 cells pre-filtration. The enucleation rates and purity of filtered reticulocytes are now added to the legend. The enucleation rates for all samples were approximately 70% and the purity of filtered reticulocytes was >99.2% in all samples.
Rather than showing a pellet of reticulocytes, the authors should show how an equal amount of erythroblasts cultured for e.g. 12 days then differentiates under explicit differentiation conditions	Cultures were started from the same starting cell number of MAM-negative and age and gender-matched control samples. The culture conditions followed a stringent protocol by which cultures were explicitly returned to the same cell density, with passaging and addition of fresh medium carried out as necessary. The proliferation graphs show the exact rate of expansion in each sample at any given day during the culture and the pellets of reticulocytes obtained after filtering of all cultures using the same filters/conditions are shown in Figure 5 (previously Figure 4) as a very visual demonstration of the differences in cell yield. These points have been emphasized in the text.
> Figure 4C shows that cells with or without EMP3 have similar differentiation profiles, and the authors suggest that it is not a delay of differentiation that causes an excess of erythroid cells after a 20 day culture. The difference in yield, according to figure 4A is 3 to 5-fold and because of the linear scale we can not interpret from where this difference originates. The authors should include a viability screen (flow cytometry?) and/or a cell division assay (using for instance CFSE) to get an idea why cell yield is different.	We have amended all the proliferation graphs to include a logarithmic scale in Figure 5 (previously Figure 4) and Supplementary Figure 6 (previously Supplementary Figure 4) in order to help visualize the time-points from which the differences in yield originate. The reviewer's suggestion of additional studies to examine the cell yield difference is certainly of interest, however, the rarity of MAM-negative material is such that we cannot do extra experiments of this nature unless more of such samples can be obtained.
> The materials and methods show two in vitro culture protocols. The legend says that 4A is the three stage culture; and the materials and methods say that phase II was changed to phase III on day 14. This fits with the sudden halt of	Two different three-stage culturing protocols were used in this study. We have now clearly separated and explained them in the Materials and Methods section. In the main text and figure legends, two cell cultures protocols are now cited as either

proliferation on day 14.	Griffiths et al. or Giarratana et al. and Flygare et al. For the majority of cell culturing experiments, including those described in Figure 5 (previously Figure 4), the Griffiths et al. three-stage protocol was used. The phases, now fully described in Materials and Methods, were phase I (days 0-11), phase II (days 11-14) and phase III (days 14-21). In this system, both MAM-negative and control samples tended to proliferate until day 16 (for example, P9 had 1.2×10^9 cells on day 14 and 2.02×10^9 cells on day 16; P10 had 8.62×10^8 cells on day 14 and 1.31×10^9 on day 16), after which they have reached a plateau and tended to differentiate. A different three-stage protocol was used for culturing of P10 sample presented in Supplementary Figure 6 (previously Supplementary Figure 3). This protocol does have slightly different stages [phase I (days 0–7), phase II (days 7-14) and phase III (days 14-18)] and the main difference is the use of dexamethasone in the medium.
I cannot believe, however, that the morphological analysis of 4C is from the same culture. It is surprising that there are already enucleated reticulocytes in presence of dexamethasone which strongly impairs enucleation. It seems that the morphological analysis of 4C stems from the Bristol type of culture which should be stated. Finally, the authors examine the interaction with CD44. At this stage it would be more clear when the authors would state that they use EMP3-negative cells. The cells may not only have lost the MAM antigen, I suppose they lack the entire protein (after all, missense mutations were found as well) Figure 5 is based on single cells. The authors may want to show additional data in the supplemental figures and/or be a bit more modest in their interpretation	Figure 5 (previously Figure 4) is showing two experiments conducted using the same Griffiths et al. culturing protocol, namely samples P9 with matched controls (coloured red) and P10 with matched controls (coloured blue), and all panels in this figure relate to these two experiments. We have amended the figure legend to include this clarification. Figure 6 (previously Figure 5) was amended and the legend now includes a clear description of test samples used (both MAM-negative and MAM-positive). Indeed, P9 MAM-negative individual shown in the figure is predicted to have lost the whole EMP3 protein because of the homozygous whole gene deletion. Supplementary Table 2 (now 3) was modified to explain in full all the mutation types found in the MAM-negative individuals. The cells shown in Figure 6 (previously Figure 5) are the representative images of in vitro cultured erythroid cells, and the legend is now stating this fact. Where appropriate, we have quantified the CD44 staining intensity for multiple cells and included the corresponding graph and statistical analysis.
In the discussion the authors try to interpret why CD44/EMP3 expression regulate erythropoiesis in vitro but not in vivo and comment on the lack of	We thank the reviewer for pointing out this in vivo compensatory mechanism. We accepted the suggestion and changed the text on page 7, lines 194-198 accordingly.

extracellular matrix in a culture flask. However, in vivo Epo levels drop when oxygen saturation is sufficient. Because lack of EMP3 is unlikely to render cells Epo independent, the Epo concentration will override the importance of EMP3 in vivo	
Reviewer #4 - expert in blood group genetics (Remarks to the Author): Thornton et al. in an overall well written manuscript provide strong genetic evidence that EMP3 is the gene responsible for expression of the previously unsolved MAM blood group, further supported by in vitro experimental evidence that the expressed EMP3 gene product is the target of anti-MAM antibodies.	No response required.
The significance of this work is that only a handful of blood group antigens remain molecularly unsolved, of which MAM is one. MAM-negative phenotypes are very rare (i.e. it is a high incidence antigen). 10 known MAM-negative individuals were studied, which substantially represents the collective MAM-negative experience, using whole exome sequencing; 5 different predicted EMP3 null-variants (1 gain-of-stop, 2 partial gene deletions, and 2 different full gene deletions) were identified. Every MAM-negative subject tested was homozygous for an EMP3 LOF variant, and nonsense variation in this gene in gnoMAD is plausibly rare.	No response required.
Thornton et al. then make a compelling case using multiple lines of evidence that EMP3 expression confers MAM antigen(s). They show that a monoclonal anti-EMP3 fails to react with MAM-negative patient RBCs but does react with reference RBCs, and that anti-MAM patient antibodies block the binding of anti-EMP3 mAb to normal RBCs.	No response required.
Compellingly, the authors are using anti-MAM containing eluates from the allosensitized MAM-negative patients – this is a real strength in the experimental design. The authors further show that shRNA EMP3 knock-down	No response required. We would like to clarify that shRNA knockdown and CRISPR/Cas9 gene editing

reduces MAM expression in the immortalized erythroid progenitor cell line BEL-A2, and that CRISPR/Cas9 disruption of the EMP3 gene in Daudi cells (a Burkitt cell line) abolished MAM expression, and that expression of EMP3 cDNA in these cells rescues anti-MAM reactivity.	knockout experiments were done using the BEL-A2 cell line, whilst the overexpression was completed in Daudi cells (a Burkitt cell line). Wild-type BEL-A2 cells express EMP3 in low levels and wild-type Daudi cells have very low levels of EMP3 expression. To clarify the experimental procedures, we have amended the text on pages 5-6, lines 149-164 and split the original Figure 3 into two new figures – Figure 3 illustrating knockout and Figure 4 illustrating overexpression.
The authors also made observations around role of EMP3 in erythropoiesis in cultured primary erythroid progenitors from 2 patients and 4 normal subjects find increased ex vivo proliferation. Pursuing the possible association with CD44, in normal RBCs they find that EMP3 and CD44 co-IP and that loss of EMP3 expression results in aberrant CD44 localization (diffuse cell surface staining and new co-localization with CD81 in vesicles) and apparent differences in CD44 abundance with differentiation (up in early stage cells, more markedly down in more mature erythroid cells).	No response required.
The authors speculate that EMP3 may be needed to stabilize or appropriately localize CD44 in the membrane, and therefore loss of EMP3 would be lead to lower CD44 expression in mature cells. Erythroid progenitor cultures from MAM-negative subjects further demonstrated increased proliferation in vitro – which follows what was previously thought to be the function of EMP3 in oncogenesis, although there is no evidence of erythrocytosis, reticulocytosis, or increased risk for malignancy in the MAM-negative individuals so the in vivo significance of this finding is unclear.	No response required.
Taken together this is a compelling paper that MAM blood group is encoded by EMP3 using multiple lines of evidence.	We would like to thank the reviewer for such a positive assessment of our work.
Comments: - The recruitment of materials from 10 MAM-negative subjects is a testament to a huge collaborative effort and the authors are to be commended	We would like to thank the reviewer for acknowledging the collaborative effort involved in recruitment of MAM-negative subjects and matched control individuals.
- Throughout, please use HGVS nomenclature to describe the genetic variants and provide the IDs for the genome and transcript references used o Please avoid the use of the word “mutation”, which is problematic because of	We have changed the description of genetic variants to HGVS nomenclature throughout the text, figures and tables and we included the genome and transcript references.

entanglements with frequency and/or function in common definitions, suggest the use of variant and its permutations	We have changed the word ‘mutation’ to ‘variation’ where it was more applicable (page 5, line 145), but we have kept the word mutation in situations where we would argue that it better expressed the rarity (frequency) or functionality (changed phenotype) of the change.
- Family information: o Please indicate subject relatedness either by coding family IDs into each unique subject identifier or by using a separate family/pedigree identifying number o Please provide the relationship to the proband in a separate column in the supplemental tables o Please define P and F subjects in the first supplemental table	We have accepted these suggestions and changed Supplementary Table 1 accordingly. The table title now reads ‘MAM-negative patient samples (coded P1-P10) and P9’s family samples (coded P9-mother, P9-father, P9-sibling) included in the study’, defining P and F subjects. We have coded family IDs into unique subject identifiers in column 1 of the table, e.g. mother of P9 individual is now coded as P9-mother. We have not expanded the table with an additional column stating relationships to the probands, because there were only three samples in this category and we hope that the new integrated subject identifiers explain the family relationships clearly. Sample P3 is listed as sibling of P2, however, P3 was also one of the MAM-negative patients in their own right.
- All subjects are homozygous for their null EMP3 variant, which is a little surprising given the rarity – this suggests either consanguinity or that there could be heterozygosity for deletions (hemizygous) o Please verify there is no evidence for loss of read depth in the WES and also note if there is heterozygosity of other regional variants o Please also comment on presence or absence of consanguinity (if not known please state) – if unknown relatedness could be assessed using the WES if this is very clearly OK with the consent/IRB and the study subjects to do such an analysis o Alternately, acknowledging some limitations of WES, a haplotype analysis could lend insights into if similar variants have arisen independently or if they appear to share a common founder allele	We have included the confirmation that there was no evidence for loss of read depth or of hemizygous gene deletions from the WES data in the notes of the Supplementary Table 2 (now 3). There is no evidence of compound heterozygosity in any of the patients’ samples, except for P9, who was a compound heterozygote for two different whole gene deletions. No informative heterozygous variants were observed in WES data, but read depths suggested homozygosity, not hemizygosity for the deletions. Consanguinity in patients’ families was unknown and not considered. Most likely there is relatedness within families, as is usually the case with rare mutations. In the manuscript, we have described the mutations, but have not commented on their ethnic distribution, which can be discerned from Supplementary Table 1. For example, all the patients with gain-of-stop mutation were of shared ethnicity (Arabic, Yemeni, Middle Eastern), whilst P5 and P8 with the same deletion were both of Turkish origin. Different mutations likely reflected regional variants.
- On page 4, the exclusion of CD44H coding variants in WES has no background	We accept there was a lack of clarity in describing the progress of the experiments

for the reader (presumably this was tested because Indian is weak and then also because of the later CD44 findings).  o CD44H is an isoform of CD44 and not a discrete gene, please indicate what WES findings were for the whole gene – it's not been determined that the CD44 part of this EMP3 story is actually restricted to CD44H o This could be in the supplement along with any other candidate gene findings that were filtered in identifying EMP3 	and we have changed the text to improve the narrative. On page 4, lines 113-115, we now state the following: 'Despite the clear serological evidence of weakened expression of CD44 on their red cells, CD44, including the erythroid isoform CD44H, showed no coding mutations (data not shown).' In real time frame, we have Sanger sequenced CD44H erythroid isoform in the MAM-negative individuals before we embarked on WES analyses, where the whole coding sequence of CD44 confirmed the original findings. We have highlighted CD44H because MAM is a red cell antigen so our interest in the CD44/EMP3 interaction was primarily directed towards red cell expression. Due to the relatively large number of unrelated MAM-negative test samples, EMP3 was the only candidate gene that passed our filtering strategy and we have now acknowledged that in the legend of Figure 1 (previously Figure 2).
 - I am glad to see the gnomAD frequencies o Please correct the numbers to accurately reflect what is currently reported in gnomAD – for example, there are 43 p.Tyr41Ter alleles out of 251,000 subjects, not 44 out of 245,818 – this is of no consequence to the authors' conclusions, but it should be corrected before publication o the authors should mention in the supplement that the annotation in gnomAD only has 4 exons while the reference transcript used elsewhere has 5 exons (I'm assuming the difference is the UTR), this is avoid confusion to readers (harmonizing with HGVS will also help avoid confusion). o The authors could also note that there were no homozygous LOFs in gnomAD o Although it is true that four subjects with EMP3 c.123C>G were detected these were not independently discovered and not related to population frequency as at least some are related, this should be considered when discussing frequency on page 5 	We thank the reviewer for the comments. We have updated all the gnomAD frequencies with the latest entries to the database and added the version used (v2.1.1). The EMP3 transcript used as a reference throughout the manuscript has been standardized and annotated following HGVS recommendations. We have added the description of the number of exons in the transcript as a footnote in Supplementary Table 2 (now 3): 'EMP3 reference sequence NM_001425.2, five exon transcript, coding exons 2-5'. Exon 1 of EMP3 is a non-coding exon. In Supplementary Table 4a, the following text has been added to reflect that the transcripts used in gnomAD and this study are identical with regards to the coding DNA: 'The gnomAD database uses the ENST00000270221.6 transcript with coding exons 2-5 identical to NM_001425.2 but with a slightly longer non-coding exon 1.' The transcript in gnomAD does include five exons as well, but the non-coding exon 1 is not included in their exome analysis and changes in this exon is not expected to alter MAM antigen expression. We have adjusted the title of Supplementary Table 3a (gnomAD frequencies, now 4a) to incorporate reviewer's comment: 'Nonsense variants in EMP3 found in the gnomAD exome dataset⁴ (no loss-of-function homozygous variants recorded).'

	We have changed the text on page 5, lines 125-128 to include reviewer's observation about selection of MAM-negative samples. The manuscript now reads: 'Although the subjects in this study were not discovered by population frequency analysis, the c.123C>G mutation was also the most common in this cohort, found in four of our propositae (two of them related).'
- The authors describe MAM in a way that implies it as being generally expressed on hematopoietic cells, but the findings in Montgomery et al. are not consistent with MAM being expressed on platelets. Please correct this.	Montgomery et al. data is not conclusive (excerpt from discussion: 'It remains unclear as to whether anti-MAM truly reacts with platelets. Results of testing the women's sera with the GTI Pak 12, the MAIPA assay, and the MACE indicate that MAM is not found on platelets. However, the results of testing eluates prepared by adsorbing the M.A.M., A.N., and F.K. sera onto and eluting them from platelets and then testing with RBCs and platelets, as well as results observed after multiple serum adsorptions with pooled platelets and human platelet concentrate indicate that anti-MAM does react with platelets.') and we have now preformed the necessary testing and included our data showing MAM expression on platelets in Supplementary Figure 5.
- Figures – please increase font sizes for readability o Figure 1 is probably useful for others that will study EMP3 in the future but I don't know if it adds much to findings here, so if there are needs in revision it could be kept but could go in the supplement. If this is kept please add the predicted N-linked glycosylation sites	The font size in figures and tables will be adjusted prior to proof-reading stage in line with any editorial guidelines. We would argue to keep Figure 1 as a part of main manuscript (now Figure 2), because we feel it illustrates well the EMP3 topology, size and closeness to the cell membrane discussed in the text. We have included the N-linked glycosylation sites on the topology model shown in panel 2a.
o Figure 2 § please increase or add more arrows to indicate gene direction in both panels § for the upper box see above comment about coverage depth and possible missed compound heterozygous deletions above – for instance, in P9 the upstream gene might not have good coverage relative to the control, but downstream (from what can be seen) looks somewhat similar § please indicate in the legend that light blue indicates UTRs	We agree with the reviewer's suggestions and we have added larger arrows in Figure 1 (previously Figure 2). We have added a comment about coverage depth in the figure legend, and commented in more detail about coverage depth and potential compound heterozygosity for different deletions in the notes column of Supplementary Table 3 (previously 2). We have indicated in the figure legend that light blue areas are UTRs and dark blue areas are coding exons of EMP3.
o Figure 3 please indicate which anti-MAM (an eluate I think) is being used, and in 3b please give AA translation described in the text in the figure	We have amended the Figure 3 legend to include a full description of the anti-MAM used, as 'anti-MAM (P9 eluate)'. We have checked the whole manuscript and made sure anti-MAM is described as an eluate.

	We have added aa translation to Figure 3b legend: ‘In clone E0.5-6, deletions NM_001425.2: c.44_48 in exon 2 and NM_001425.2: c.329_377 in exon 5 were close to the location of guide DNAs and both would have introduced a reading frame shift and aberrant termination of the protein translation (p.Ile15AsnfsTer40 and p.Ala115ValfsTer176, respectively).’
o Figure 4 there should be more clarity on the experiment that resulted in the cell pellet picture (is this after culture of a normalized seeded plate?)	We have expanded the Figure 5 (previously Figure 4) legend and the Materials and Methods section (page 22, lines 677-686 and lines 694-695) to address the reviewer’s comment. We now explain how all the cultures were assessed under normalized feeding regime, counted daily and adjusted to the same cell density as necessary, resulting in different volumes of MAM-negative and control samples. Furthermore, all cells cultured for 21 days were filtered on day 21 using the same leucofilters and filtration method resulting in a very visual demonstration of the differences in cell yield. The enucleation rates and purity of filtered reticulocytes are now added to the legend.
o Figure 5b it looks like there is a big difference in CD81 staining in MAM-negative cells but this is not discussed by the authors ... I mention this because it seems noteworthy as another change of loss of EMP3, if this is representative, and CD81 also has a proliferative story § In speculating on mechanism, another role for EMP3 may be appropriate intracellular trafficking, particularly given the intracellular CD81/CD44 findings	Indeed, we have only speculated that in the absence of the stabilising effect of EMP3 in MAM-negative cells, CD44 may also be lost via a different mechanism involving plasma membrane vesiculation. Our data is consistent with CD81 being associated as an accessory with CD44 in vesicles, but we have not studied intracellular trafficking of CD44, which will be a subject for future work.
o The experiments in supplemental figure 2 are compelling and I think should be in the body of the main paper – my summary of why is that this is the CRISPR/Cas9 KO of EMP3 in Daudi cells with loss of EMP3 mAb staining showing it is KO, then overexpression by transfection of EMP3 and gain of anti-MAM seroreactivity with a good EMP3 nonsense control – this is the clincher experiment they have the right gene – knock out then rescue using a very good reagent (the human eluate with anti-MAM)	We accept the reviewer’s suggestion and to that aim we have moved the overexpression experiments to a separate figure in the main text (now Figure 4). We have also expanded the narrative on page 4 and 5 in order to remove any confusion about the set of experiments involving BEL-A2 and Daudi cells. The knockdown and knockout experiments have been achieved in BEL-A2 cells which express low levels of EMP3, whilst overexpression experiment was completed in Daudi cells, which have no native EMP3. We have used human anti-MAM eluate to show the direct link with EMP3, but also because it gave a far superior result to commercially available anti-EMP3 MABs.
o Supplemental figure 4 § please show the IN weak results § in the main text and methods this figure is referred to as an IP experiment	We have added Indian (CD44) results as new Supplementary Table 2. Immunoprecipitation data and subsequent proteomics study were omitted in error and are now included in a new Supplementary Figure 7 (to replace and expand

with EMP3 and CD44 but in the figure itself looks like a regular blot – please revise to indicate exactly what was done and clarify the conclusions	previous Supplementary Figure 4a) and Supplementary Proteomics Data File 1. We have amended our discussion accordingly (page 7, lines 201-207).
- Supplemental table 4 and PNGase F o Page 15 describes RBC treatment with papain and PNGase F, and the results are seemingly in Supplemental table 4, but this is a confusing table – is everything in this result treated with papain/PNGase F? § Please clearly show the results of the treated and untreated cells. § I am very interested in clarifying the PNGase F results because treatment with PNGase F has been reported (in AABB meeting abstract form) to result in ablation of anti-MAM activity on multiple cells and collapse of the smeary bands on anti-MAM immunoblots to discrete bands – I think from this I would have expected some loss of anti-MAM eluate reactivity with PNGaseF. Although there are limitations in that the AABB abstract findings it is a rare piece of data associated with this blood group,so for the purposes of examining this as a blood group it is important to clarify what are the PNGase F findings in this context	We agree with reviewer’s comment about a confusing table and we have changed Supplementary Table 4 and replaced it with two new tables, Supplementary Table 6a and 6b, therefore clarifying the results of serology testing. The tables now describe serological tests with human anti-MAM and monoclonal anti-EMP3 with two MAM-negative patients and two MAM-positive controls (6a) and incubation of MAM-positive red cells with human anti-MAM blocking the binding of murine monoclonal anti-EMP3 (6b). Papain+PNGaseF two stage treatment is now indicated in the tables. One possible explanation of discrepancy with Lee et al. findings may be that in the original findings PNGaseF could have been contaminated with a protease, because many early commercial and in-house preps were at that time (DJ Anstee, personal communication). Alternatively, the sizes of anti-MAM-reactive red cell components reported by us concur with those of Montgomery et al. (2000), who described a sharply migrating band of approximately 18,000kDa and a broadly migrating component with a leading edge of approximately 23,000 kDa. Of significance is that Lee et al. do not report the 18,000 kDa band. The EMP3 protein is of the expected size of 18,429 kDa (Uniprot database) and the 18,000 kDa component identified by anti-MAM is likely to be non-glycosylated MAM/EMP3. As EMP3 has got two N-glycosylation sites, the larger broadly migrating component identified by anti-MAM is the heterogeneously N-glycosylated MAM/EMP3 protein. The presence of both glycosylated and non-glycosylated protein is also seen for RBC aquaporin-1 (Smith et al. 1994). The Montgomery results were obtained on large 10% non-reduced gels (F Spring, personal communication) which gave superior separation of components. Lee at al. used 10% non-reduced gels, where the 18,000 kDa band was absent as it either formed larger aggregates or it migrated at the gel dye front. By serology, PNGaseF digestion (with or without papain) did not abrogate reactivity with anti-MAM or anti-EMP3, although Lee et al. reported results to the contrary. However, poor reactivity of anti-EMP3 MAbs suggest that the position of the MAM epitope is on the extended loop 1 on the molecule, close to cell surface, and therefore it would not be surprising if extended PNGaseF treatment gave the same results as the Lee et al. abstract.

- Supplemental table 5 please also include the detection of MAM on cells of different lineages from the MAM blood group publications in addition to cancer lineages, which given that malignancy does not seem to be part of the MAM blood group or erythroid story right now and normal tissue expression patterns are of interest	We agree with the comment and we have added new Supplementary Figure 2 depicting EMP3 expression levels in normal tissues and a selection of cell lines.
- If the authors need room in revision, they could summarize the PMP22 story more succinctly.	We thank the reviewer for the suggestion, but we would prefer to keep the PMP22 story since the word count is within Nature Communications limit.

Reviewers' Comments:

Reviewer #1:

Remarks to the Author:

In the paper by Thornton et. al, the authors evaluate the association of EMP3 in erythroid proliferation as well as in regulation the MAM negative phenotype. In this study, the authors examine 10 MAM negative individuals and find that all 10 show that EMP3 may be a candidate to help regulate MAM expression. The paper strives to connect the association of EMP3 with MAM and CD44, and in this revision, this feat is better accomplished. However, while in the revision the authors show a clear pattern of MAM regulation by EMP3, the lack of mechanism makes the paper still a bit premature.

Specific concerns:

1. It's difficult to appreciate some of the significance of the results. For example:

A. The links between CD44, CD81 and MAM/EMP3 are still correlative and at times, the results seem disparate. For example, for Fig.6C, the authors write, "We can speculate that in the absence of the stabilising effect of EMP3 in MAM-negative cells, CD44 may also be lost via a different mechanism involving plasma membrane vesiculation." However, in supplemental fig. 3, changes in EMP3 in BEL-A2 cells did not change CD44 mRNA or expression levels while MAM expression did change.

B. The authors write that EMP3 acts as a suppressor of proliferation in normal erythropoiesis (Figure 5 and Supplementary Figure 6). "However, MAM-negative individuals do not exhibit any apparent erythrocytosis or reticulocytosis (data not shown)." While their explanation is plausible, the mechanism(s) of action then remain unknown.

Minor Points:

1. Supplemental Table 6b: It's unclear to me how this scale was determined and what the data means given that the authors write in Supplementary Table 4, "commercially available monoclonal anti-EMP3 proved to be poor reagents and a low level of staining of KO cells was observed with both flow cytometry and immunofluorescence methods. We suspect this was due to cross-reactivity with the other members of the EMP family of proteins."

2. Supplementary Figure 7B i and ii (between the RBC and IP portions) look like the blots are spliced together. While this is not a concern, if true, perhaps it would be worthwhile to put a dashed line where the blot was cut?

3. The statement "CD81 regulates the expression of CD19 at the surface of B lymphocytes²¹" seems out of place to me (p7). Moreover, the relationship between CD81 and EMP3 did not seem to connect well.

Reviewer #2:

Remarks to the Author:

Thornton et al., report on the identification of the epithelial membrane protein 3 (EMP3) as the antigen corresponding to the MAM blood group. In the MAM-negative individuals, CD44 is weakly expressed.

There is the possibility that CD44 is not the only protein affected by mutations in the EMP3 gene in MAM-negative patients. Yet, the authors decided to stress the potential interaction between CD44 and EMP3 and a large part of the paper is focused on CD44. Unfortunately, the data provided on CD44 are descriptive.

In response to the reviewers, the authors argue that the potential mechanistic link between CD44 and EMP3 is subject of future work.

However, since the authors extensively speculate on the potential role of CD44 in the effect of the EMP3 mutations, at least one mechanistic hint should be advanced.

For example, following their hypothesis that the regulation of CD44 by EMP3 has an effect on

MAM-negative proerythroblasts proliferation (lines 225-231), the manipulation of CD44 might result in a partial rescue of the phenotype in EMP3 negative erythroblasts?

In addition, the authors should be cautious with the concept of protein "stabilization" by EMP3 (line 236). Indeed, since the level of CD44 is increased in the erythroblasts of MAM-negative individuals, CD44 seems to be stabilized by the absence of EMP3 in this context. Nevertheless, it is understandable that the possible complexity of this molecular mechanism could require efforts beyond the scope of this paper.

Furthermore, the authors claim a direct association between CD44 and EMP3. Do they mean that the molecules directly interact? Since co-IPs cannot rule out interactions mediated by other proteins, the authors should clarify what they mean by "direct association".

Reviewer #3:

Remarks to the Author:

I considered the revised manuscript and read the rebuttal of the authors.

The authors responded to all concerns raised, and the manuscript has greatly improved.

I only have few minor comments

1)

121 Nonsense mutations in
122 EMP3 are rare (Supplementary Tables 4a, 4b) but c.123C>G (p.Tyr41Ter) is
123 by far the most commonly encountered in the Genome Aggregation Database
124 (gnomAD), where it is present in 43 of 251,000 alleles (0.017%).

English is not my mother tongue. To me the connection "but" is weird and confusing because the first sentence is not a contradiction to the second sentence. To me "whereas" would be a more clear fusion of two independent remarks on two different types of mutations.

2)

133 Members of this family have been shown
134 to be involved in diverse cellular (mostly membrane) functions, ranging from
135 membrane organisation to regulation of tight junctions¹¹, but generally forming
136 barriers in the membrane.

Tight junctions are membrane barriers, and are formed by membrane organization. I suggest to strike part of the sentence as indicated

3)

246 Despite the putative cancer
247 association of EMP3, we have no evidence that EMP3null individuals have an
248 increased risk of cancer.

The numbers of EMP3null individuals are far too small to say something about a cancer risk ?!!

4)

256 The absence of some or all of these

257 extracellular signals may explain why the enhanced erythroid proliferation we
258 observe in cultures from MAM-negative individuals does not appear to be
259 replicated in vivo.

The authors have added a note on feed back control at an earlier stage of the manuscript

194 Since in vivo erythropoietin levels decrease when oxygen saturation is
195 sufficient and the lack of EMP3 is unlikely to render cells erythropoietin
196 independent, we can speculate that the erythropoietin concentration will
197 override the importance of EMP3 regulation in vivo, or other compensatory
198 mechanisms may be at play.

This also applies here, may even better apply here?

At least, the lines 256-259 could be slightly adapted to fit the notion of Epo feedback control

Marieke von Lindern

Reviewer #4:

Remarks to the Author:

Herein is a revised manuscript that reflects additional work and improvements by the authors. My specific critiques have been addressed and the thoughtful nature of the authors' rebuttals is appreciated. I have the following comments related to the blood group MAM and genetics:

-- The investigation of MAM expression on platelets is relevant new data (with appropriate negative control) that has the potential to provide additional clarity on the potential pathologies of anti-MAM, particularly in light of the authors' comments on associated thrombocytopenia.

-- the substance of the discussion on page 16 in the rebuttal giving an overview of what was known of MAM previously would be of benefit to readers to have in the manuscript ; the table is improved

Minor:

-- The authors' use of "mutation" is not in line with HGVS (HGVS points to Richards et al, 2015 to use the neutral term variant) – as long as the actual variants being discussed are clear, if the authors prefer older terminology I will defer to editors.

-- the comment that the filters used in exome studies presumed unrelatedness is not ideal, as elsewhere in the authors speculate on cryptic relatedness as a possible explanation for homozygosity - however, this almost certainly has little bearing on the overall conclusions as these authors only found one gene, there are no unexplained families, and the gene is experimental validated.

Author response to reviewer and editor comments on manuscript NCOMMS-18-37307B

Reviewer comment:	Author response:
Reviewer #1 (Remarks to the Author): In the paper by Thornton et. al, the authors evaluate the association of EMP3 in erythroid proliferation as well as in regulation the MAM negative phenotype. In this study, the authors examine 10 MAM negative individuals and find that all 10 show that EMP3 may be a candidate to help regulate MAM expression. The paper strives to connect the association of EMP3 with MAM and CD44, and in this revision, this feat is better accomplished. However, while in the revision the authors show a clear pattern of MAM regulation by EMP3, the lack of mechanism makes the paper still a bit premature.	This study aimed and succeeded to elucidate the molecular and genetic basis of the MAM-negative blood group phenotype in all known MAM negative individuals at the time of the study (10 individuals). We have used three different approaches to prove the direct mechanism of regulation of MAM by EMP3, i.e. EMP3 knock-down, knock-out and overexpression. In addition, we have provided novel and clear data about how EMP3 affects erythropoiesis in vitro and how EMP3 relates to CD44 in red cell membranes. In this revision, we have included new experiments to better explain and start unravelling the observed association of EMP3/MAM and CD44; however, as the Reviewer comments later in the review, fully elucidating the mechanism of CD44 interaction with EMP3/MAM would comprise a substantial amount of extra work for the purposes of this paper. The Reviewer suggests there is no clear mechanism of MAM regulation by EMP3 shown here. We would argue that we have clearly shown the mechanism of genetic control of MAM phenotype by EMP3, which is explained clearer and in more detail in this version of the manuscript as it is the main focus of the paper.
Specific concerns: 1. It's difficult to appreciate some of the significance of the results. For example: A. The links between CD44, CD81 and MAM/EMP3 are still correlative and at times, the results seem disparate. For example, for Fig.6C, the authors write, "We can speculate that in the absence of the stabilising effect of EMP3 in MAM-negative cells, CD44 may also be lost via a different mechanism involving plasma membrane vesiculation." However, in supplemental fig. 3, changes in EMP3 in BEL-A2 cells did not change CD44 mRNA or expression levels while MAM expression did change.	To address the specific concerns of the Reviewer, we have included additional data from two new experiments, a MAIEA assay and a CD44 manipulation by CRISPR/Cas9 CD44 knock-out in BEL-A2 cell line. We have outlined the links between MAM/EMP3 and CD44 more thoroughly, and explained that CD81 was only tested by confocal microscopy amongst other tetraspanins and proteins of interest. The two examples the Reviewer suggest are disparate, in fact support our supposition that MAM/EMP3 stabilises CD44 in red cell membranes, and we have endeavored to provide new evidence and alter the manuscript to clarify this. Our new data shows that MAM/EMP3 expression does not require CD44 presence, whereas CD44 expression on cell surfaces is affected by the lack of MAM/EMP3. MAM/EMP3 is clearly expressed in the absence of CD44, which aligns with our initial proposition that EMP3 stabilises CD44 in the plasma membrane, but the two

	are not associated during biosynthesis. To that point, we have highlighted the Reviewer’s observation about no change of CD44 mRNA levels in the EMP3 knock-down BEL-A2 cell line. It follows from this that MAM/EMP3 levels would not be expected to be affected in the CD44 KO BEL-A2 lines, and indeed they were not. We have also added new results from the MAIEA assay which clearly support association of the two proteins in the plasma membrane.
B. The authors write that EMP3 acts as a suppressor of proliferation in normal erythropoiesis (Figure 5 and Supplementary Figure 6). “However, MAM-negative individuals do not exhibit any apparent erythrocytosis or reticulocytosis (data not shown).” While their explanation is plausible, the mechanism(s) of action then remain unknown.	The Reviewer correctly highlights the absence of the mechanism of action in vivo; however, the link between EMP3 and proliferation is well-established in the literature and has been discussed in the paper. Further elucidation of the mechanism is desirable, but the volume of work required would suggest that it should be regarded in a separate future study.
Minor Points: 1. Supplemental Table 6b: It’s unclear to me how this scale was determined and what the data means given that the authors write in Supplementary Table 4, “commercially available monoclonal anti-EMP3 proved to be poor reagents and a low level of staining of KO cells was observed with both flow cytometry and immunofluorescence methods. We suspect this was due to cross-reactivity with the other members of the EMP family of proteins.”	We have expanded the explanation of serology and blocking experiments to address the Reviewer’s concern. Indeed, the monoclonal anti-EMP3 reagents were very poor in many methods, including the testing of untreated cells by serology. The enzyme treatment required to enable antibody binding was possible when treating intact red cells, but this enzyme treatment could not be withstood by BEL-A2 cells that had been manipulated and therefore were too fragile. Although the anti-EMP3 reactivity observed in these tests appears robust, the enzyme treatment of the cells was a difficult process. We have amended Supplementary Table 6b and the relevant methods section to indicate that enzyme treated cells were used for the blocking tests and clarified the need for enzyme treatment in the main text (page 6, lines 156-162). The grading scale of serological reactivity is the standardized 0-5+ scale. This is now added to the Methods, serological testing section, page 12, lines 345-346.
2. Supplementary Figure 7B i and ii (between the RBC and IP portions) look like the blots are spliced together. While this is not a concern, if true, perhaps it would be worthwhile to put a dashed line where the blot was cut?	We apologise to the Reviewer for this oversight, the SOURCE data file submitted did not contain our blotting images. This data has now been added and clearly shows where blots have been spliced together, rather than adding a dashed line to the final image.

3. The statement “CD81 regulates the expression of CD19 at the surface of B lymphocytes²¹” seems out of place to me (p7). Moreover, the relationship between CD81 and EMP3 did not seem to connect well.	The Reviewer is correct in noting the lack of clarity in this statement. The statement “CD81 regulates the expression of CD19 at the surface of B lymphocytes” was intended as an analogy, and our apologies that this was not clear. We have edited this paragraph and also added further information about the results of confocal experiments undertaken and the clear differences in CD44 distribution in MAM-negative cells. Please see page 7, lines 216-217 and page 8, lines 220-229.
Reviewer #2 (Remarks to the Author): Thorton et al., report on the identification of the epithelial membrane protein 3 (EMP3) as the antigen corresponding to the MAM blood group. In the MAM-negative individuals, CD44 is weakly expressed. There is the possibility that CD44 is not the only protein affected by mutations in the EMP3 gene in MAM-negative patients. Yet, the authors decided to stress the potential interaction between CD44 and EMP3 and a large part of the paper is focused on CD44. Unfortunately, the data provided on CD44 are descriptive. In response to the reviewers, the authors argue that the potential mechanistic link between CD44 and EMP3 is subject of future work. However, since the authors extensively speculate on the potential role of CD44 in the effect of the EMP3 mutations, at least one mechanistic hint should be advanced. For example, following their hypothesis that the regulation of CD44 by EMP3 has an effect on MAM-negative proerythroblasts proliferation (lines 225-231), the manipulation of CD44 might result in a partial rescue of the phenotype in EMP3 negative erythroblasts?	We agree with the Reviewer’s comment that the reasoning behind our investigation of CD44 expression in MAM-negative individuals was perhaps not fully explained. We have now expanded this section (page 4, lines 105-116) to clearly describe how CD44 was the only known red cell surface protein with weakened expression in MAM-negative individuals. We have now added the results of the extensive serological investigation of high prevalence antigens encompassing all blood group system carriers (Supplementary Table 2) and also the MAIEA assay (Supplementary Table 4), which is designed to locate blood group antigens on specific red cell membrane proteins. Both experiments indicated a clear connection of MAM with CD44, and no other tested red cell membrane protein. As suggested by the Reviewer, we have advanced one experimental approach to elucidate the mechanistic link between MAM/EMP3 and CD44 and have now shown the close physical interaction between the two by using the sensitive ELISA-based MAIEA assay. Furthermore, we have manipulated the BEL-A2 cell line (also used for our EMP3 knock-out) to produce CD44 knock-out and explore the expression of MAM/EMP3 in the absence of CD44 (Supplementary Figure 9). We have now shown that in a CD44 KO BEL-A2 line, the lack of CD44 does not alter MAM expression. These two experiments provide clear insight into the interaction between the two proteins in red cell membranes and we have altered the manuscript to emphasize this point. There was no evidence of growth advantage in CD44 knock-out BEL-A2 cell line when compared to control treated with an empty vector; suggesting that an external ligand may be needed for its function. However, to investigate the growth patterns further and compare them directly to growth advantage observed in MAM-negative individuals, we would need a CD44 human null (knock-out)

	phenotype. We believe that when saying “partial rescue of the phenotype” the Reviewer is suggesting the growth advantage that we have reported due to EMP3 mutations. Whilst we are lucky to have an EMP3 human knock-out, there is no known CD44 human KO to replicate the growth experiments in a comparable way.
In addition, the authors should be cautious with the concept of protein “stabilization” by EMP3 (line 236). Indeed, since the level of CD44 is increased in the erythroblasts of MAM-negative individuals, CD44 seems to be stabilized by the absence of EMP3 in this context. Nevertheless, it is understandable that the possible complexity of this molecular mechanism could require efforts beyond the scope of this paper.	We have shown the CD44 expression in ex vivo cultured erythroid cells in both MAM-negative and control samples, where in both phenotypes, CD44 is present in early differentiation stages and is lost from cell membranes as cells mature (as seen in Supplementary Figure 8). The greater loss of CD44 in MAM-negative cells is evident, and we tried to clarify the impact of the altered CD44 distribution in MAM-negative erythroblasts (in cleavage furrow and in membrane vesicles) and the resultant gross reduction of CD44 expression in mature erythrocytes and cultured reticulocytes (line 236), i.e. although they have more CD44, they also lose more. We have now included the evidence of a direct physical interaction of MAM/EMP3 and CD44 in red cell membranes (MAIEA assay, page 4, lines 105-116 and Supplementary Table 4), but we agree with the Reviewer that the full elucidation of this complex interaction is beyond the scope of this paper.
Furthermore, the authors claim a direct association between CD44 and EMP3. Do they mean that the molecules directly interact? Since co-IPs cannot rule out interactions mediated by other proteins, the authors should clarify what they mean by “direct association”.	We agree with the Reviewer that co-IP cannot rule out interactions between EMP3 and other proteins apart from CD44. We have, however, added new data to the manuscript to clarify why the direct association between CD44 and EMP3 is evident. (MAIEA assay, please see the comment above).
Reviewer #3 (Remarks to the Author): I considered the revised manuscript and read the rebuttal of the authors. The authors responded to all concerns raised, and the manuscript has greatly improved. I only have few minor comments.	We thank the Reviewer for their kind comment. We thank the reviewer for the comment and we have removed “but” and replaced

1) Nonsense mutations in EMP3 are rare (Supplementary Tables 4a, 4b) but c.123C>G (p.Tyr41Ter) is by far the most commonly encountered in the Genome Aggregation Database (gnomAD), where it is present in 43 of 251,000 alleles (0.017%). English is not my mother tongue. To me the connection “but” is weird and confusing because the first sentence is not a contradiction to the second sentence. To me “whereas” would be a more clear fusion of two independent remarks on two different types of mutations.	with a semi-colon to clarify that in both cases we are discussing inactivating (nonsense) mutations.
2) 133 Members of this family have been shown 134 to be involved in diverse cellular (mostly membrane) functions, ranging from (135) membrane organisation to regulation of tight junctions¹¹, but generally forming (136) barriers in the membrane. Tight junctions are membrane barriers, and are formed by membrane organization. I suggest to strike part of the sentence as indicated.	Thank you, we have changed the text as suggested by the Reviewer (page 5, lines 71-72).
3) 246 Despite the putative cancer 247 association of EMP3, we have no evidence that EMP3null individuals have an 248 increased risk of cancer. The numbers of EMP3null individuals are far too small to say something about a cancer risk ?!!	Despite the small number of EMP3_{null} individuals, we are stating the facts as seen by the available data and we have no intention of claiming that there is no risk. There are other rare blood group null phenotypes that are associated with known pathological phenotypes and even premature death, e.g. CD151 (RAPH), and this fact is also based on only a small number of known cases.
4) The absence of some or all of these extracellular signals may explain why the enhanced erythroid proliferation we observe in cultures from MAM-negative individuals does not appear to be replicated in vivo. The authors have added a note on feed back control at an earlier stage of the manuscript	We’d like to thank the Reviewer for this observation and we have moved the paragraph in question to the more appropriate place as suggested (page 9, lines 273-280).

Since in vivo erythropoietin levels decrease when oxygen saturation is sufficient and the lack of EMP3 is unlikely to render cells erythropoietin independent, we can speculate that the erythropoietin concentration will override the importance of EMP3 regulation in vivo, or other compensatory mechanisms may be at play. This also applies here, may even better apply here? At least, the lines 256-259 could be slightly adapted to fit the notion of Epo feedback control.	
Reviewer #4 (Remarks to the Author): Herein is a revised manuscript that reflects additional work and improvements by the authors. My specific critiques have been addressed and the thoughtful nature of the authors' rebuttals is appreciated.	We would like to thank the Reviewer for their positive comments.
I have the following comments related to the blood group MAM and genetics: -- The investigation of MAM expression on platelets is relevant new data (with appropriate negative control) that has the potential to provide additional clarity on the potential pathologies of anti-MAM, particularly in light of the authors' comments on associated thrombocytopenia.	Thank you.
-- the substance of the discussion on page 16 in the rebuttal giving an overview of what was known of MAM previously would be of benefit to readers to have in the manuscript ; the table is improved	Thank you.
Minor: -- The authors' use of "mutation" is not in line with HGVS (HGVS points to Richards et al, 2015 to use the neutral term variant) – as long as the actual variants being discussed are clear, if the authors prefer older terminology I will defer to editors.	We would still argue that the word mutation is more accurate when describing changes of the genetic origin, but we are aware of the current changes in nomenclature and are happy to change if required by editors.
-- the comment that the filters used in exome studies presumed unrelatedness is not ideal, as elsewhere in the authors speculate on cryptic relatedness as a possible explanation for homozygosity - however, this almost certainly has little	Thank you; no change required.

bearing on the overall conclusions as these authors only found one gene, there are no unexplained families, and the gene is experimental validated.

Editor remarks to the Author:

All reviewers raised additional points that would need to be addressed. We are interested in the possibility of publishing your study in Nature Communications, but would like to consider your response to these concerns in the form of a revised manuscript before we make a final decision on publication.

In particular, we would ask a revised manuscript to include experiments manipulating CD44 expression, as requested by reviewer #2 at this and the previous round of review, in order to address the concerns of both reviewer #1 and reviewer #2 regarding the level of mechanistic insights provided in your study. Inclusion of such data will be required for further consideration of your manuscript. All other concerns raised by our reviewers should also be addressed.

We therefore invite you to revise and resubmit your manuscript, taking into account the points raised. Please highlight all changes in the manuscript text file.

We are committed to providing a fair and constructive peer-review process. Do not hesitate to contact us if you wish to discuss the revision in more detail or if there are specific requests from the reviewers that you believe are technically impossible or unlikely to yield a meaningful outcome.

Author response:

We have revised the manuscript to clarify it and include new data that provide better explanation as to why CD44 was a protein of interest from the very beginning of this study (extensive serological investigation of MAM phenotype; page 4, lines 101-116 and Supplementary Table 2, Supplementary Table 3 and Supplementary Figure 1).

We have also addressed the critique by the reviewers #1 and #2 by including two new experiments that address the question of the level of mechanistic insights of MAM/EMP3 relationship provided in our original study. The MAIEA assay is a sensitive ELISA-based method developed for locating red cell antigens on specific red cell membrane proteins and our new MAIEA data suggest a direct physical relationship between MAM/EMP3 with CD44. This data strengthens the co-IP data that was already presented in the manuscript.

We have also followed the reviewers' recommendation to manipulate CD44, and for that purpose we have used CRISPR/Cas9 gene editing in the BEL-A2 cell line to abolish the expression of CD44 and observe its effect of MAM/EMP3 expression. We have shown that CD44 is not needed for the expression of MAM/EMP3, suggesting a direct and undisputable regulation of MAM phenotype by *EMP3*. This new data is supportive of our supposition that MAM/EMP3 is needed for the expression of CD44 in red cell membranes, but not for the biosynthesis of the protein, as suggested by Supplementary Figure 2a and Supplementary Figure 8b. We have followed the manuscript checklist and have reorganized the text to follow the introduction-results-discussion format. In the process, the original writing was minimally changed, but the formatting has allowed us to even more clearly support the main focus of our paper, i.e. that MAM antigen is encoded by *EMP3*. Their relationship with CD44 is now hopefully better explained through our new data from MAIEA and CD44 KO experiments, whilst unravelling the complex nature of this relationship will be a subject of a further study.

We have highlighted all the new data and discussion in the manuscript, whilst the original text moved only because of the format editing is not highlighted, to ensure

	true changes were easily identified. We are happy to highlight format changes if required.

Reviewers' Comments:

Reviewer #1:

Remarks to the Author:

In this revised paper by Thornton et. al, the authors work to establish an interaction between EMP3 and the MAM phenotype. Using cultured erythroid progenitor cells from MAM-negative individuals, their data suggests that EMP3 associates with CD44 and may have a regulatory role in erythropoiesis and control of cell production. While the revision continues to be constrained by the lack of quality anti-EMP3 reagents and is largely correlative, the authors have largely addressed the concerns of most reviewers.

Reviewer #2:

Remarks to the Author:

In this new version, the authors provide additional data supporting an association between CD44 and EMP3. In addition, a CD44KO was achieved in BEL-A2 cells. This CD44 KO does not impact on EMP3 expression. Altogether, the experiments that were requested in the previous reviewing were performed. Despite all the remaining questions, this paper can now be published in Nature Comm.

Reviewers' Comments:

Reviewer #1:

Remarks to the Author:

In this revised paper by Thornton et. al, the authors work to establish an interaction between EMP3 and the MAM phenotype. Using cultured erythroid progenitor cells from MAM-negative individuals, their data suggests that EMP3 associates with CD44 and may have a regulatory role in erythropoiesis and control of cell production. While the revision continues to be constrained by the lack of quality anti-EMP3 reagents and is largely correlative, the authors have largely addressed the concerns of most reviewers.

Reviewer #2:

Remarks to the Author:

In this new version, the authors provide additional data supporting an association between CD44 and EMP3. In addition, a CD44KO was achieved in BEL-A2 cells. This CD44 KO does not impact on EMP3 expression. Altogether, the experiments that were requested in the previous reviewing were performed. Despite all the remaining questions, this paper can now be published in Nature Comm.

We thank the reviewers for their comments.